# Q-Learning with Adjoint Matching

**Qiyang Li**
UC Berkeley
qcli@berkeley.edu

**Sergey Levine**
UC Berkeley
svlevine@berkeley.edu

## Abstract

We propose Q-learning with Adjoint Matching (QAM), a novel TD-based reinforcement learning (RL) algorithm that tackles a long-standing challenge in continuous-action RL: efficient optimization of an expressive diffusion or flow-matching policy with respect to a parameterized Q-function. Effective optimization requires exploiting the first-order information of the critic, but it is challenging to do so for flow or diffusion policies because direct gradient-based optimization via backpropagation through their multi-step denoising process is numerically unstable. Existing methods work around this either by only using the value and discarding the gradient information, or by relying on approximations that sacrifice policy expressivity or bias the learned policy. QAM sidesteps both of these challenges by leveraging adjoint matching, a recently proposed technique in generative modeling, which transforms the critic's action gradient to form a step-wise objective function that is free from unstable backpropagation, while providing an unbiased, expressive policy at the optimum. Combined with temporal-difference backup for critic learning, QAM consistently outperforms prior approaches on hard, sparse reward tasks in both offline and offline-to-online RL. **Code:** github.com/ColinQiyangLi/qam

## 1 Introduction

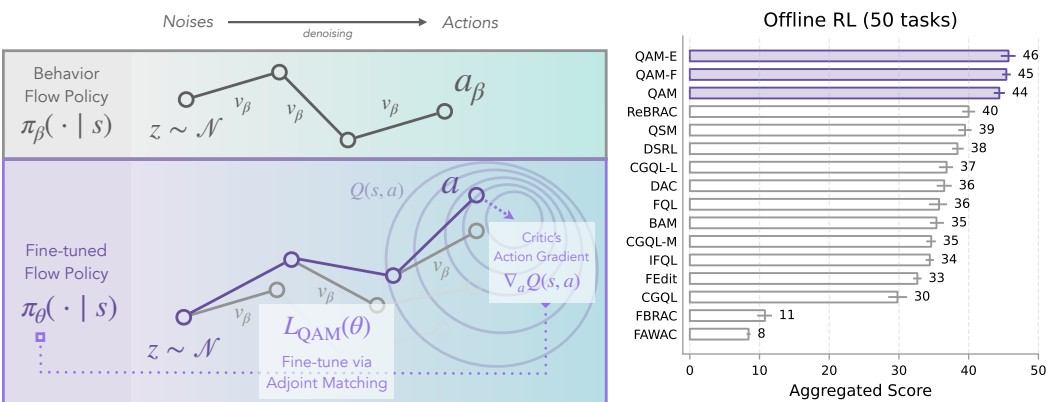

Figure 1: **QAM: Q-learning with Adjoint Matching.** *Left:* QAM uses adjoint matching (Domingo-Enrich et al., 2025) that leverages the critic's action gradient directly to fine-tune a flow policy towards the optimal behavior-constrained policy: $\pi_\theta(\cdot \mid s) \propto \pi_\beta(\cdot \mid s)e^{Q(s,\cdot)}$. *Right:* Aggregated score for offline RL on 50 OGBench (Park et al., 2025a) tasks.

A long-standing tension in continuous-action reinforcement learning (RL) especially in the offline/offline-to-online setting, is between policy expressivity and optimization tractability with respect to a critic (*i.e.*, $Q(s, a)$). Simple policies, such as single-step Gaussian policies, are easy to train, since they can directly leverage the critic's action gradient (*i.e.*, $\nabla_a Q(s, a)$) via the reparameterization trick (Haarnoja et al., 2018). This optimization tractability, however, often comes at the cost of expressivity. Some of the most expressive policy classes today, such as flow or diffusion policies, generate actions through a multi-step denoising process. While this allows them to represent complex,

multi-modal action distributions, leveraging the action gradient requires backpropagation through the entire denoising process, which often leads to instability (Park et al., 2025c). Prior work has therefore resorted to either (1) discarding the critic's action gradient entirely and only using its value (Ren et al., 2025; Zhang et al., 2025; McAllister et al., 2025), or (2) distilling expressive, multi-step flow policies into one-step noise-conditioned approximations (Park et al., 2025c). The former sacrifices learning efficiency and often under-performs methods that use the critic's action gradient (Park et al., 2024; 2025c), while the latter compromises expressivity. This raises a question: can we somehow keep the full expressivity of flow policies while incorporating the critic's action gradient directly into the denoising process without backpropagation instability?

One might be tempted to directly apply the critic's action gradient to intermediate noisy actions within the denoising process, as in diffusion classifier guidance (with the critic function being the classifier) (Dhariwal & Nichol, 2021). Intuitively, this blends two generative process together: one that generates a behavior action distribution, and another that hill-climbs the critic to maximize action value. While this approach bypasses the backpropagation instability and retains full policy expressivity, it relies on the assumption that the critic's gradient at a noisy action is a good proxy for its gradient at the corresponding denoised action. In practice, this assumption often breaks down: when the offline dataset has limited action coverage, the critic is well-trained only on a narrow distribution of noiseless actions, rendering its gradients unreliable for intermediate noisy actions that are out of distribution.

We propose **Q-learning with Adjoint Matching (QAM)**, a new RL algorithm that applies adjoint matching (Domingo-Enrich et al., 2025), a recently developed technique in generative modeling, to policy optimization. QAM leverages the critic's action gradient for training flow policies to maximize returns subject to a prior constraint (*e.g.*, behavior constraint) (Figure 1). In general, such a constrained optimization problem on a flow model can be formulated as a stochastic optimal control (SOC) objective, which can be solved by using the continuous adjoint method (Pontryagin et al., 1962). However, this standard formulation has the same loss landscape as directly backpropagating through the SOC objective, causing instability. Instead, we leverage a modified objective from Domingo-Enrich et al. (2025) that admits the same optimal solution, but does not suffer from the instability challenge. At a high level, the critic's gradient at noiseless actions is directly transformed by a flow model constructed from the prior, independent from the possibly ill-conditioned flow model that is being optimized, to construct unbiased gradient estimates for optimizing the state-conditioned velocity field at intermediate denoising steps. This allows the flow policy's velocity field to align directly with the optimal state-conditioned velocity field implied by the critic and the prior, without direct and potentially unstable backpropagation, while preserving the full expressivity of multi-step flow models. By combining this policy extraction procedure with a standard temporal-difference (TD) backup for critic learning, QAM enables the flow policy to efficiently converge to the optimal policy subject to the prior constraint. In contrast, approximation methods that rely on the critic's gradients at noisy intermediate actions lack such convergence guarantees.

Our main contribution is a new TD-based RL algorithm that leverages adjoint matching to perform policy extraction effectively on a critic. Unlikely prior Q-learning methods with flow-matching that rely on approximations or throwing away the action gradient of the critic altogether, our algorithm directly uses the gradient to form an objective that at convergence recovers the optimal behavior-regularized policy. We conduct a comprehensive empirical study comparing policy extraction methods for flow/diffusion policies, including recent approaches and novel baselines, and show that QAM consistently achieves strong performance across both offline RL and offline-to-online RL benchmarks.

## 2 RELATED WORK

**RL with diffusion and flow policies.** Diffusion and flow policies have been explored in both policy gradient methods (Ren et al., 2025) and actor-critic methods (Fang et al., 2025b; Kang et al., 2023; Chen et al., 2024c;a; Lu et al., 2023b; Ding et al., 2024b; Wang et al., 2023; He et al., 2023; Ding & Jin, 2024; Ada et al., 2024; Zhang et al., 2024; Hansen-Estruch et al., 2023). The key challenge of leveraging diffusion/flow policies in TD-based RL methods is to optimize these policies against the critic function (*i.e.*, $Q(s, a)$). Prior work can be largely put into three categories based on how the value function is used:

(1) *Post-processing* approaches refine the action distribution from a base diffusion/flow policy with rejection sampling based on the critic value (Hansen-Estruch et al., 2023; Mark et al., 2025; Li et al.,

2025; Dong et al., 2025), or using additional gradient steps to hill climb the critic (Mark et al., 2025) (*i.e.*, $a_t \leftarrow a_t + \nabla_a Q(s, a)$). These approaches often reliably improve the quality of extracted policy but at the expense of additional computation during evaluation or even training (*i.e.*, rejection sampling for value backup target (Li et al., 2025; Dong et al., 2025)). Alternatively, one may train a residual policy that modifies a base behavior policy in either the noise space (Singh et al., 2021; Wagenmaker et al., 2025) or in the action space directly (Yuan et al., 2025; Ankile et al., 2025a;b; Dong et al., 2025).

(2) *Backprop-based* approaches perform direct backpropagation through both the critic and the policy (Wang et al., 2023; He et al., 2023; Ding & Jin, 2023; Zhang et al., 2024; Park et al., 2025c; Espinosa-Dice et al., 2025; Chen et al., 2025). While this is the most-straightforward implementation-wise, it requires backpropagation through the diffusion/flow policy's denoising process which has been observed to be unstable (Park et al., 2025c), or instead learns a distilled policy (Ding & Jin, 2023; Chen et al., 2024b; Park et al., 2025c; Espinosa-Dice et al., 2025; Chen et al., 2025), at the expense of policy expressivity.

(3) *Intermediate fine-tuning* approaches, which our method also belongs to, mitigate the need of the stability/expressivity trade-off in backprop-based approaches by leveraging the critic to construct an objective that provides direct step-wise supervision to the intermediate denoising process (Psenka et al., 2024a; Fang et al., 2025b; Ding et al., 2024a; Li et al., 2024b; Frans et al., 2025; Zhang et al., 2025; Ma et al., 2025; Koirala & Fleming, 2025). While these approaches remove the need for backpropagation through the denoising process completely, the challenge lies in carefully crafting the step-wise objective that does not introduce additional biases and learning instability. Compared to prior methods that either rely on approximations (Lu et al., 2023a; Fang et al., 2025b) that do not provide theoretical guarantees (see more discussions in Section C) or directly throwing away the critic's action gradient (and use its value instead) (Ding et al., 2024a; Zhang et al., 2025; Ma et al., 2025; Koirala & Fleming, 2025), we leverage adjoint matching (Domingo-Enrich et al., 2025) which allows us to use the critic's action gradient directly to construct an direct step-wise objective for our flow policy that recovers the optimal prior regularized policy at the optimum of the objective.

**Offline-to-online reinforcement learning** methods focus on leveraging offline RL to first pre-train on an offline dataset, and then use the pretrained policy and value function(s) as initialization to accelerate online RL (Xie et al., 2021; Song et al., 2023; Lee et al., 2022; Agarwal et al., 2022; Zhang et al., 2023; Zheng et al., 2023; Ball et al., 2023; Nakamoto et al., 2024; Li et al., 2024a; Wilcoxson et al., 2025; Zhou et al., 2025). While it is possible to skip the offline pre-training phase altogether and use online RL methods directly by treating the offline dataset as additional off-policy data that is pre-loaded into the replay buffer (Lee et al., 2022; Song et al., 2023; Ball et al., 2023), these methods often under-perform the methods that leverage explicit offline pre-training, especially on more challenging tasks (Nakamoto et al., 2024; Park et al., 2025c). Our method also operates in this regime where we first perform offline RL pre-training and then perform online fine-tuning from the offline pre-trained initialization. In addition, we follow a common design in prior work where the same offline RL objective is used for both offline pre-training and online fine-tuning (Kostrikov et al., 2022; Fujimoto & Gu, 2021; Tarasov et al., 2023; Park et al., 2025c). While we focus on evaluating our method in the offline-to-online RL setting, the idea of using adjoint matching to train an expressive flow policy can be applied to other settings such as online RL.

For additional related work on diffusion and flow-matching with guidance, see Section C.

## 3 PRELIMINARIES

**Reinforcement learning and problem setup.** We consider a Markov Decision Process (MDP), $\mathcal{M} = (\mathcal{S}, \mathcal{A}, P, \gamma, R, \mu)$, where $\mathcal{S}$ is the state space, $\mathcal{A} = \mathbb{R}^A$ ($A \in \mathbb{Z}^+$) is the action space, $P : \mathcal{S} \times \mathcal{A} \to \Delta_{\mathcal{S}}$ is the transition function, $\gamma \in [0, 1)$ is the discount factor, $R : \mathcal{S} \times \mathbb{R}^A \to \mathbb{R}$ is the reward function, and $\mu \in \Delta_{\mathcal{A}}$ is the initial state distribution. We have access to a dataset $D$ consisting of a set of transitions $D = \{(s_i, a_i, s_i', r_i)\}_{i=1}^{|D|}$, where $s' \sim P(\cdot \mid s, a)$ and $r = R(s, a)$. Our first goal (*offline RL*) is to learn a policy $\pi_\theta : \mathcal{S} \to \Delta_{\mathcal{A}}$ from $D$ that maximizes its expected discounted return,

$$U_\pi = \mathbb{E}_{s_0 \sim \mu, s_{k+1} \sim P(\cdot|s_k, a_k), a_k \sim \pi(\cdot|s_k)} \left[ \sum_{k=0}^{\infty} \gamma^k R(s_k, a_k) \right]. \tag{1}$$

The second goal (*offline-to-online RL*) is to fine-tune the offline pre-trained policy $\pi_\theta$ by continuously interacting with the MDP through trajectory episodes with a task/environment dependent maximum

episode length of $H$ (*i.e.*, the maximum number of time steps before the agent is reset to $\mu$). The central challenge of offline-to-online RL is to maximally leverage the behavior prior in $D$ to learn as sample-efficiently (high return with few environment interactions) as possible online.

**Flow-matching generative model.** A flow model uses a time-variant velocity field $v : \mathbb{R}^d \times [0, 1] \to \mathbb{R}^d$ to estimate the marginal distribution of a denoising process from noise, $X_0 = \mathcal{N}(0, I_d)$, to data, $X_1 = D$, at each intermediate time $t \in [0, 1]$:

$$X_t = (1 - t)X_0 + tX_1. \tag{2}$$

In particular, the flow model approximates the intermediate $X_t$ via an ordinary differential equation (ODE) starting from the noise: $X_0 = \mathcal{N}$:

$$\mathrm{d}\hat{X}_t = f(\hat{X}_t, t)\mathrm{d}t. \tag{3}$$

Flow models are typically trained with a *flow matching* objective (Liu et al., 2023):

$$L_{\mathrm{FM}}(\theta) = \mathbb{E}_{t \sim \mathcal{U}[0,1], x_0 \sim \mathcal{N}, x_1 \sim D} \left[ \| f_\theta((1 - t)x_0 + tx_1, t) - x_1 + x_0 \|_2^2 \right], \tag{4}$$

where any optimal velocity field, $v_{\theta^\star}$, results in $\hat{X}_t$ where its marginal distribution $p_f(x_t)$ exactly recovers the marginal distribution of the original denoising process $X_t$, $p_D(x_t)$, for each $t \in [0, 1]$ (Lipman et al., 2024). Furthermore, one may use the Fokker-Planck equations to construct a family of stochastic differential equations (SDE) that admits the same marginals as well:

$$\mathrm{d}\hat{X}_t = \left( f(\hat{X}_t, t) + \frac{\sigma_t^2 t}{2(1 - t)} \left( f(\hat{X}_t, t) + X_t/t \right) \right) \mathrm{d}t + \sigma_t \mathrm{d}B_t \tag{5}$$

with $B_t$ being a Brownian motion and $\sigma_t > 0$ being any noise schedule.

**Adjoint matching** is a technique developed by Domingo-Enrich et al. (2025) with the goal of modifying a base flow-matching generative model $f_\beta$ such that the resulting flow model $f_\theta$ generates the following *tilt* distribution:

$$p_\theta(X_1) \propto p_\beta(X_1)e^{Q(X_1)} \tag{6}$$

where $p_\theta$ is the tilt distribution induced by $f_\theta$ and $p_\beta$ is the base distribution induced by $f_\beta$, and $Q : \mathbb{R}^d \to \mathbb{R}$ is any value function that up-weights or down-weights the probability of each example in the domain $\mathbb{R}^d$. Domingo-Enrich et al. (2025) uses a marginal-preserving SDE with a 'memoryless' noise schedule (*i.e.*, $X_0$ and $X_1$ are independent), $\sigma_t = \sqrt{2(1 - t)/t}$:[1]

$$\mathrm{d}X_t = (2f(X_t, t) - X_t/t)\,\mathrm{d}t + \sqrt{2(1 - t)/t}\mathrm{d}B_t, \tag{7}$$

where the minimizer of the following stochastic optimal control equation (with $X_t$ sampling from the joint distribution defined by the SDE in Equation (7)),

$$L(\theta) = \mathbb{E}_{\boldsymbol{X}=\{X_t\}_t} \left[ \int_0^1 \left( \frac{2}{\sigma_t^2} \| f_\theta(X_t, t) - f_\beta(X_t, t) \|_2^2 \right) \mathrm{d}t - Q(X_1) \right] \tag{8}$$

gives the correct marginal *tilt* distribution for $X_1$ (Domingo-Enrich et al., 2024):

$$p(X_1) \propto p_\beta(X_1)e^{Q(X_1)}. \tag{9}$$

Let the adjoint state be the gradient of the tilt function applied at the denoised $X_1$:

$$g(\boldsymbol{X}, t) = \nabla_{X_t} \left[ \int_t^1 \left( \frac{2}{\sigma_{t'}^2} \| f_\theta(X_{t'}, t') - f_\beta(X_{t'}, t') \|_2^2 \right) \mathrm{d}t' - Q(X_1) \right], \tag{10}$$

where $\boldsymbol{X} = \{X_t\}_t$ is the random variable that represents the SDE trajectory, and $g$ satisfies

$$\frac{\mathrm{d}g(\boldsymbol{X}, t)}{\mathrm{d}t} = -\nabla_{X_t} \left[ 2f_\theta(X_t, t) - X_t/t \right] g(\boldsymbol{X}, t) - \frac{2}{\sigma_t^2} \nabla_{X_t} \| f_\theta(X_t, t) - f_\beta(X_t, t) \|_2^2, \tag{11}$$

---

[1]The adjoint-matching paper (Domingo-Enrich et al., 2025) uses a more general framework with $X_t = \beta_t X_0 + \alpha_t X_1$. For simplicity, we assume the special case where $\alpha_t = t$, $\beta_t = 1 - t$, and $\sigma_t = \sqrt{2(1 - t)/t}$.

with the boundary condition $g(\boldsymbol{X}, t = 1) = -\nabla_{X_1} Q(X_1)$. We can compute the adjoint states by stepping through the reverse ODE (which can be efficiently computed with the vector-Jacobian product (VJP) in most modern deep learning frameworks). Then, it can be shown that the stochastic optimal control loss in Equation (8) is equivalent to the 'basic' adjoint matching objective below:

$$L_{\text{BAM}}(\theta) = \mathbb{E}_{\boldsymbol{X}} \left[ \int_0^1 \|2(f_\theta(X_t, t) - f_\beta(X_t, t))/\sigma_t + \sigma_t g(\boldsymbol{X}, t)\|_2^2 dt \right]. \tag{12}$$

The optimal $f_\theta$ coincides with the optimal solution in the original SOC equation (Equation (8)), which gives the correct marginal distribution of $X_1$ as a result. However, the objective is equivalent to the objective used in the continuous adjoint method (Pontryagin et al., 1962) with its gradient equal to that of backpropagation through the denoising process. Instead, Domingo-Enrich et al. (2025) derive the 'lean' adjoint state where all the terms in the adjoint state that are zero at the optimum are removed from the original adjoint state. The 'lean' adjoint state satisfies the following ODE:

$$d\tilde{g}(\boldsymbol{X}, t) = -\nabla_{X_t} \left[ 2f_\beta(X_t, t) - X_t/t \right] \tilde{g}(\boldsymbol{X}, t) dt, \tag{13}$$

with the same boundary condition $\tilde{g}(\boldsymbol{X}, 1) = -\nabla_{X_1} Q(X_1)$.

Note that computing the 'lean' adjoint state only requires the base flow model $f_\beta(X_t, t)$ and no longer needs to use $f_\theta(X_t, t)$ as needed in either the basic adjoint matching objective (Equation (12)) or naive backpropagation through the denoising process. The resulting adjoint matching objective is

$$L_{\text{AM}}(\theta) = \mathbb{E}_{\boldsymbol{X}} \left[ \int_0^1 \|2(f_\theta(X_t, t) - f_\beta(X_t, t))/\sigma_t + \sigma_t \tilde{g}(\boldsymbol{X}, t)\|_2^2 dt \right], \tag{14}$$

where again $\boldsymbol{X}$ is sampled from the marginal preserving SDE in Equation (7). Because the terms omitted in the 'lean' adjoint state are zero at the optimum, and thus do not change the optimal solution for $f_\theta$. Thus, the optimal solution for the adjoint matching gives the correct tilt distribution.

## 4 Q-LEARNING WITH ADJOINT MATCHING (QAM)

In this section, we describe in detail how our method leverages adjoint matching to directly align the flow policy to prior regularized optimal policy without suffering from backpropagation instability.

To start with, we first define the optimal policy that we want to learn as the solution of the best policy under the standard KL behavior constraint:

$$\arg\max_\pi \mathbb{E}_{a \sim \pi(\cdot|s)}[Q(s, a)] \quad \text{s.t.} \quad D_{\text{KL}}(\pi \| \pi_\beta) \leq \epsilon(s). \tag{15}$$

or equivalently, for an appropriate $\tau(s)$, Nair et al. (2020) show that

$$\pi^\star(a \mid s) \propto \pi_\beta(a \mid s) e^{\tau(s) Q_\phi(s, a)} \tag{16}$$

where $\tau : \mathcal{S} \to \mathbb{R}^+$ is the inverse temperature coefficient that controls the strength of the behavior constraint at each state.

We approximate the behavior policy using a flow-matching behavior policy, $f_\beta : \mathcal{S} \times \mathbb{R}^A \times [0, 1] \to \mathbb{R}^A$ that is optimized with the standard flow-matching objective:

$$L_{\text{FM}}(\beta) = \mathbb{E}_{(s,a) \sim D, t \sim [0,1], z \sim \mathcal{N}} \left[ \|f_\beta(s, (1 - t)z + ta, t) - a + z\|_2^2 \right] \tag{17}$$

We then parameterize our approximation of the optimal policy as another flow model $f_\theta : \mathcal{S} \times \mathbb{R}^A \times [0, 1] \to \mathbb{R}^A$ and solve the following SOC equation:

$$L(\theta) = \mathbb{E}_{s \sim D, a_t} \left[ \int_0^1 \frac{2}{\sigma_t^2} \|f_\theta(s, a_t, t) - f_\beta(s, a_t, t)\|_2^2 - \tau(s) Q_\phi(s, a_1) dt \right], \tag{18}$$

where $a_t$ is defined by the following 'memory-less' SDE (e.g., $a_0$ is independent from $a_1$):

$$da_t = (2f_\theta(s, a_t, t) - a_t/t) dt + \sqrt{2(1 - t)/t} dB_t. \tag{19}$$

Similar to the derivation by Domingo-Enrich et al. (2025), the memory-less property allows us to directly conclude that the SOC equation has the optimum at

$$\pi_\theta(\cdot \mid s) \propto \pi_\beta(\cdot \mid s)e^{\tau(s)Q_\phi(s,a)} \tag{20}$$

where $\pi_\theta(\cdot \mid s)$ and $\pi_\beta(\cdot \mid s)$ are the corresponding action distributions defined by $f_\theta$ and $f_\beta$.

However, directly solving the SOC equation involves backpropagation through time that introduces additional instability. To circumvent this issue, we use the adjoint matching objective proposed by Domingo-Enrich et al. (2025) (Equation (14)) to construct a similar objective for policy optimization in our case:

$$L_{\mathrm{AM}}(\theta) = \mathbb{E}_{s\sim D,\{a_t\}_t}\left[\int_0^1 \|2(f_\theta(s,a_t,t) - f_\beta(s,a_t,t))/\sigma_t + \sigma_t \tilde{g}_t\|_2^2 \mathrm{d}t\right] \tag{21}$$

where $\tilde{g}_t$ is the 'lean' adjoint state defined by a reverse ODE constructed from $a_t$'s:

$$\mathrm{d}\tilde{g}_t = -\nabla_{a_t}\left[2f_\beta(s,a_t,t) - a_t/t\right]\tilde{g}_t \mathrm{d}t. \tag{22}$$

Unlike the original SOC objective (Equation (18)) from which calculating the gradient requires backpropagating through an SDE, which suffers from stability challenges, the adjoint matching objective is constructed without backpropagation. Instead, it uses the behavior velocity field $f_\beta$ to calculate the 'lean' adjoint states $\{\tilde{g}_t\}_t$ through a series of VJPs for every SDE trajectory $\{a_t\}_t$, which are then used to form a squared loss in the adjoint matching objective. Mathematically, backpropagation can also be interpreted as calculating the adjoint states through a series of VJPs, with the key distinction that the VJPs are computed under the flow model that is being optimized (*i.e.*, $f_\theta$). This is an important distinction because for direct backpropagation, any ill-conditioned action gradient in $f_\theta$ (*i.e.*, $\nabla_a f_\theta(s,a,t)$) would compound over the entire denoising process, contributing to the 'ill-condition-ness' of the overall gradient to the parameter $\theta$, which can in turn destabilize the whole optimization process. In contrast, in adjoint matching, the action gradient of $f_\theta$ has no contribution to the overall gradient to $\theta$, which allows the optimization to be much more stable.

Finally, we combine the policy optimization with the standard TD-learning objective:

$$L(\phi) = \mathbb{E}_{s,a,s',r\sim D}\left[(Q(s,a) - r - \gamma Q_{\bar{\phi}}(s',a'))\right], \quad a' \leftarrow \mathrm{ODE}(f_\theta(s',\cdot,\cdot), a_0' \sim \mathcal{N}) \tag{23}$$

where $\mathrm{ODE}(f_\theta(s',\cdot,\cdot), a_0') := \int_0^1 f_\theta(s',a_t',t)\mathrm{d}t$ (sampling an action sample from the velocity field $f_\theta(s',\cdot,\cdot)$) and $\bar{\phi}$ is the exponential moving average of $\phi$ with a time-constant of $\lambda = 0.005$ (*i.e.*, $\bar{\phi}_{i+1} \leftarrow (1-\lambda)\bar{\phi}_i + \lambda\phi_i$ for each training step $i$).

**Practical considerations.** In practice, following Domingo-Enrich et al. (2025), we solve both the SDE and the reverse ODE with discrete approximation and a fixed step size of $h = 1/T$, where $T$ is the number of discretization steps. In particular, with $a_0 \sim \mathcal{N}$ and $z_t \sim \mathcal{N}, \forall t \in \{0, h, \cdots (T-1)h\}$, the forward SDE process is approximated by

$$a_{t+h} \leftarrow a_t + h \cdot (2f_\theta(s,a_t,t) - a_t/t) + \sqrt{2h(1-t)/t}z_t. \tag{24}$$

We set the boundary condition as $\tilde{g}_1 = -\tau\nabla_{a_1}Q_\phi(s,a_1)$, where we use a state-independent inverse temperature coefficient $\tau$ to modulate the influence of the prior $\pi_\beta$ and we additionally clip the magnitude of the parameter gradient element-wise by 1 for numerical stability. The backward adjoint state calculation process is then approximated by

$$\tilde{g}_{t-h} \leftarrow \tilde{g}_t + h \cdot \mathrm{VJP}(\nabla_{a_t}(2f_\beta(s,a_t,t) - a_t/t), \tilde{g}_t), \tag{25}$$

with $\mathrm{VJP}(\nabla_{\boldsymbol{y}}b(\boldsymbol{y}), \boldsymbol{x}) = [\nabla_{\boldsymbol{y}}b(\boldsymbol{y})]\boldsymbol{x}$ being the vector-Jacobian product and it can be practically implemented by carrying the 'gradient' $x$ with backpropagation through $f$. For the critic, we use an ensemble of $K = 10$ critic functions $\phi^1, \cdots, \phi^K$ and use the pessimistic target value backup following Fang et al. (2025a) (originally inspired by Ghasemipour et al. (2022)). The loss function for each $\phi^j, j \in \{1,2,\cdots,K\}$ is

$$L(\phi^j) = \left(Q_{\phi^j}(s,a) - r - \gamma\left[\bar{Q}_{\mathrm{mean}}(s',a') - \rho\bar{Q}_{\mathrm{std}}(s',a')\right]\right)^2, \tag{26}$$

where $\bar{Q}_{\mathrm{mean}}(s',a') := \frac{1}{K}\sum_k Q_{\bar{\phi}^k}(s',a')$, $\bar{Q}_{\mathrm{std}}(s',a') = \frac{1}{K}\sqrt{\sum_k(Q_{\bar{\phi}^k}(s',a') - \bar{Q}_{\mathrm{mean}}(s',a'))^2}$, and $a'$ is the action sampled from the fine-tuned flow model: $f_\theta(s,\cdot,\cdot)$. For all our experiments,

we do not use a separate training process for $f_\beta$ and instead training it at the same time as $f_\theta$ and $Q_\phi$, following Park et al. (2025c); Li et al. (2025). For all our loss functions, the transition tuple $(s, a, s', r)$ is drawn from $D$ uniformly. During offline training, $D$ is the offline data. During online fine-tuning, $D$ is combination of the offline and online replay buffer data without any re-weighting. A pseudocode of our algorithm is available in Algorithm 1 in the Appendix.

**Theoretical guarantees.** As our algorithm builds off from Domingo-Enrich et al. (2025), we can directly extend their theoretical results to our setting (see in Section G)—as long as the loss function $L_{\text{AM}}$ is optimized to convergence (*i.e.*, with $\partial L / \partial f_\theta = 0$), the learned policy *coincides* with the optimal behavior-constrained policy (*i.e.*, $\pi^\star(a \mid s) \propto \pi_\beta(a \mid s) \exp(\tau Q(s, a))$ in Equation (16)).

**Expanding the constraint set beyond the KL behavior constraint.** While our method, QAM, is guaranteed to converge to the optimal behavior-constrained policy (*i.e.*, $\propto \pi_\beta \exp(Q)$), it can struggle to represent any policy that has a support mismatch with the behavior policy. For tasks where the optimal actions have extremely low probability under the behavior distribution, our algorithm may have trouble to represent an optimal policy for these tasks. In practice, we often find it to be beneficial to relax the behavior constraint in QAM to allow actions that are 'close' (in the action space) to the behavior actions. This amounts to optimizing the policy under the following constraint:

$$\arg\max_{\pi, \tilde{\pi}} \mathbb{E}_{a \sim \pi(\cdot|s)}[Q(s, a)] \quad \text{s.t.} \quad D_{\text{KL}}(\tilde{\pi} \parallel \pi_\beta) \leq \epsilon(s), W_q(\pi, \tilde{\pi}) \leq \sigma_a, \quad (27)$$

where $W$ is the $q$-Wasserstein distance between $\pi, \tilde{\pi}$ under some metric $d : \mathcal{A} \times \mathcal{A} \mapsto \mathbb{R}_0^+$, and $\sigma_a$ is some constant that controls the strength of the constraint:

$$W_q(\pi(\cdot \mid s), \tilde{\pi}(\cdot \mid s)) := \inf_{c \in \mathcal{C}(\pi, \tilde{\pi})} \left[ \mathbb{E}_{a, \tilde{a} \sim c(a, \tilde{a}|s)} \left[ d(a, \tilde{a})^q \right]^{1/q} \right], \quad (28)$$

where $\mathcal{C}(\pi, \tilde{\pi})$ is the set of all couplings between $\pi$ and $\tilde{\pi}$. With such design, $\pi$ not only respects KL behavior constraint, but can also represent actions that are 'close' (in the geometry of the action space) to the behavior policy. While directly optimizing this objective is intractable, we approximate it by using the solution from QAM (*i.e.*, $\pi_\theta$) to approximate the optimal $\tilde{\pi}$, allowing us to transpose the problem into a behavior-constrained policy optimization problem with $\pi_\theta$ being the base policy:

$$\pi^\star \approx \pi_\omega = \arg\max_\pi \mathbb{E}_{a \sim \pi(\cdot|s)}[Q(s, a)] \quad \text{s.t.} \quad W_q(\pi(a \mid s), \pi_\theta(a \mid s)) \leq \sigma_a. \quad (29)$$

In practice, we explore two practical variants for optimizing $\pi$ with different choices of $q$ and $d$ and approximations.

*The first variant,* `QAM-FQL`: For our first variant, we use $q = 2$ and $d(a, \tilde{a}) = \|a - \tilde{a}\|_2$, which allows us to directly use a prior flow RL method, FQL (Park et al., 2025c). In particular, we learn a 1-step noise-conditioned policy, $\pi_\omega$ (as represented by $\mu_\omega(s, z) : \mathcal{S} \times \mathbb{R}^A \mapsto \mathbb{R}^A$), and optimize the following objective (with $\alpha$ being a tunable BC coefficient),

$$L_{\text{QAM-FQL}}(\omega) = \mathbb{E}_{z \sim \mathcal{N}} \left[ -Q_\phi(s, \mu_\omega(s, z)) + \alpha \| \mu_\omega(s, z) - \text{ODE}(f_\theta(s, \cdot, \cdot), z) \|_2^2 \right]. \quad (30)$$

As a direct application of the theoretical result in Park et al. (2025c), the second term in the equation above is an upper-bound on squared Wasserstein distance $W_2^2(\pi_\omega, \pi_\theta)$ between the 1-step noise-conditioned policy $\pi_\omega$ and the optimal behavior-constrained policy obtained via QAM, $\pi_\theta$. This is exactly the desired constraint (*i.e.*, $W_2^2(\pi_\omega, \pi_\theta) \leq \epsilon_W$). We then use this 1-step policy $\pi_\omega$ both to interact with the environment and to compute value targets (*i.e.*, $\bar{Q}(s', a' = \mu(s, z \sim \mathcal{N}))$). Intuitively, the 1-step policy is optimized to remain close to the QAM-fine-tuned flow policy while also maximizing the action value under the current critic. We call this first variant of our method `QAM-FQL`.

*The second variant,* `QAM-EDIT`: Our second variant uses $q = 1$ and $d(a, \tilde{a}) = \|a - \tilde{a}\|_\infty$. To implement this, we learn an edit policy $\pi_\omega(\Delta_a \mid s, a)$ that predicts an action edit $\Delta_a$ to modify the action output from $\pi_\theta$ by at most $\sigma_a$ in the $L_\infty$ distance (*i.e.*, $\|\Delta_a\|_\infty \leq \sigma_a$). This ensures that the Wasserstein distance constraint (*i.e.*, $W_1(\pi_\omega, \pi_\theta) \leq \sigma_a$) is enforced by construction. In practice, we train an edit policy (Dong et al., 2025), $\pi_\omega(\cdot \mid s, \tilde{a})$, where it first predicts a Gaussian distribution, then squashed by a tanh function to between $-1$ and $1$, and finally rescaled to be between $-\sigma_a$ and $\sigma_a$ as the action edit. To avoid the degenerate case where the edit policy always outputs the boundary action edits (*i.e.*, $-\sigma_a$ or $\sigma_a$), we also include the standard entropy term from Haarnoja et al. (2018) with the automatic entropy tuning trick such that the policy's entropy is close to a constant value of $\mathcal{H}(\pi_\omega) = \mathcal{H}_{\text{target}} = -A/2$.

$$L_{\text{QAM-EDIT}}(\omega) = \mathbb{E}_{\Delta_a \sim \pi_\omega(\cdot|s,a)} \left[ -Q_\phi(s, \Delta_a + a) + \eta \log \pi_\omega(\Delta_a \mid s, a) \right],$$
$$L(\eta) = \eta \left[ \mathbb{E}_{\Delta_a \sim \pi_\omega(\cdot|s,a)} \left[ \log \pi_\omega(\Delta_a \mid s, a) \right] - \mathcal{H}_{\text{target}} \right],$$
$$(31)$$

where $a \sim \pi_\theta(\cdot \mid s)$ for both losses and again $\Delta_a$ is restricted to be $\|\Delta_a\|_\infty \leq \sigma_a$. Similarly to our first variant, we use $\pi_\omega$ both to interact with the environment and to compute value targets. A potential side benefit of the entropy term is that it can also encourage action diversity that can be helpful for online exploration and fine-tuning. As we will show in our experiments, this QAM variant not only excels in offline RL but also fine-tunes effectively online.

## 5 EXPERIMENTS

We conduct experiments to evaluate the effectiveness of our method on a range of long-horizon, sparse-reward domains and compare it against a set of representative baselines.

**Domains and datasets.** We consider 10 domains (5 tasks each) from OGBench (Park et al., 2025a): `scene`, `puzzle-3x3` (p33), `puzzle-4x4` (p44), `cube-double` (c2), `cube-triple` (c3), `cube-quadruple` (c4), `humanoidmaze-medium` (hm), `humanoidmaze-large` (hl), `antmaze-large` (al), and `antmaze-giant` (ag). For `antmaze-*` and `humanoidmaze-*`, we use the default `navigate` datasets. For `scene`, `puzzle-*`, and `cube-*`, we use the default `play` datasets except for c4 and p44 where we use the larger 100M-size dataset from Park et al. (2025b), and use the sparse reward definition for {p33, p44, scene}, following Li et al. (2025). In addition, for all {cube-*, scene-*, p33-*, p44-*} domains, we follow Li et al. (2025) to learn action chunking policies with an action chunking size of $h = 5$. Action chunking policies output high-dimensional actions that exhibits a much more complex behavior distribution, where the policy extraction becomes critical. Since our approach primarily focuses on the policy extraction aspect, these domains make an ideal testbed us to compare our method to prior work. See Section D for more details on these domains.

**Comparisons.** We carefully select 13 representative, strong baselines that can be roughly categorized into the following 5 categories— **(1) Gaussian:** ReBRAC (Tarasov et al., 2023), **(2) Backprop:** FBRAC (Park et al., 2025c) (backprop through the flow policy's denoising step directly), FQL (backprop through a 1-step distilled policy) (Park et al., 2025c); **(3) Advantage-weighted:** FAWAC (Park et al., 2025c) (advantage weighted actor critic, AWAC (Nair et al., 2020), with flow policy); **(4) Guidance with the critic's action gradient:** DAC (Fang et al., 2025b), QSM (Psenka et al., 2024a), and CGQL/CGQL-MSE/CGQL-Linex (three variants of classifier guidance-based methods inspired by Dhariwal & Nichol (2021)); **(5) Post-processing-based:** DSRL (Wagenmaker et al., 2025), FEdit (flow + Gaussian edit policy from Dong et al. (2025)), and IFQL (flow counterpart of IDQL (Hansen-Estruch et al., 2023)). For offline-to-online evaluations we additionally compare with RLPD (Ball et al., 2023). Finally, we compare with BAM, a direct ablation from our method QAM where we use the 'basic' adjoint matching (and hence the abbreviation BAM) objective in Equation (12) instead of the adjoint matching objective in Equation (14), and keep the rest of the implementation exactly the same. We categorize it as a "backprop" method because its gradient is equivalent to that of backpropagating through the memory-less SDE as we discuss above in Section 3.

Among them, RLPD does not employ any behavior constraint, so we directly train from scratch online. To make the comparison fair, we use $K = 10$ critic networks, pessimistic value backup with $\rho = 0.5$ (except on `humanoidmaze-large` where we find $\rho = 0$ to work better), no best-of-N sampling (*i.e.*, $N = 1$) for both our methods and our baselines except for IFQL where best-of-N is used for policy extraction. See Section E for detailed description and implementation detail for each of the baselines. We also include the domain-specific hyperparameters for each baseline in Section F.

## 6 RESULTS

In this section, we present our experimental results to answer the following three questions:

**(Q1) How effective is our method for offline RL?**

Table 1 reports the offline RL performance across 10 different domains (50 tasks in total). QAM outperforms all prior methods with an aggregated score of 44. In particular, among all the baselines, FAWAC also converges to the optimal behavior-constrained distribution (Equation (16) but does not leverage the action gradient of the critic, resulting in a much worse aggregated score of 8; BAM also leverages the SOC objective but optimizes it with backpropagation through the SDE chain (without using the 'lean' adjoint), leading to a worse aggregated score of 35. It is worth noting that FBRAC also leverages backpropagation through time (BPTT) similar to BAM, but performs much worse than BAM. The main distinction between BAM and FBRAC is that BAM leverages the SOC formulation

| | | al (5 tasks) | ag (5 tasks) | hm (5 tasks) | hl (5 tasks) | scene (5 tasks) | p33 (5 tasks) | p44 (5 tasks) | c2 (5 tasks) | c3 (5 tasks) | c4 (5 tasks) | all (50 tasks) |
|---|---|---|---|---|---|---|---|---|---|---|---|---|
| GAUSSIAN | ReBRAC | **94** [94,95] | **57** [53,60] | 69 [65,74] | 17 [15,19] | 65 [61,69] | 79 [73,84] | 0 [0,0] | 9 [9,10] | 1 [0,1] | 9 [6,11] | 40 [39,41] |
| BACKPROP | FBRAC | 2 [1,4] | 0 [0,0] | 39 [37,41] | 0 [0,0] | 50 [43,57] | 0 [0,1] | 15 [12,19] | 0 [0,0] | 0 [0,1] | 0 [0,0] | 11 [10,12] |
| | BAM | 84 [62,85] | 1 [0,2] | 60 [58,62] | 5 [4,8] | **98** [97,99] | 56 [48,64] | 0 [0,0] | 47 [44,50] | 3 [2,5] | 0 [0,0] | 35 [34,36] |
| | FQL | 76 [72,79] | 0 [0,0] | 68 [63,73] | 9 [7,11] | 78 [77,80] | 70 [60,76] | 5 [3,7] | 46 [43,49] | 3 [2,4] | 2 [1,5] | 36 [34,37] |
| ADV. WEIGHTED | FAWAC | 17 [15,19] | 0 [0,0] | 24 [22,26] | 0 [0,0] | 38 [35,41] | 3 [2,3] | 0 [0,0] | 2 [2,2] | 0 [0,0] | 0 [0,0] | 8 [6,9] |
| GUIDANCE | CGQL | 76 [73,80] | 0 [0,2] | 60 [57,62] | 5 [4,5] | 38 [36,40] | 48 [40,55] | 24 [16,33] | 38 [36,41] | **8** [7,9] | 0 [0,0] | 30 [29,31] |
| | CGQL-M | 71 [69,73] | 4 [1,8] | 42 [40,43] | 6 [3,8] | 74 [72,76] | **100** [100,100] | 0 [0,0] | 41 [39,43] | **8** [7,9] | 1 [0,1] | 35 [34,35] |
| | CGQL-L | 65 [62,67] | 3 [1,6] | 62 [57,66] | 6 [5,8] | 88 [85,91] | 90 [83,96] | 0 [0,0] | 45 [43,47] | **8** [7,9] | 0 [0,1] | 37 [36,38] |
| | DAC | 88 [86,90] | 16 [11,20] | **83** [81,85] | 0 [0,0] | 68 [65,70] | 68 [62,75] | 0 [0,0] | 35 [33,36] | 5 [3,6] | 3 [1,5] | 36 [35,38] |
| | QSM | 90 [88,92] | 24 [19,29] | 82 [79,84] | 6 [5,7] | 78 [77,80] | 57 [52,63] | 0 [0,0] | 33 [31,34] | 6 [5,6] | 19 [18,19] | 39 [38,40] |
| POST-PROCESSING | DSRL | 61 [56,66] | 3 [1,4] | 53 [48,58] | 3 [2,5] | **99** [99,100] | 87 [82,92] | 0 [0,0] | **74** [72,76] | 1 [1,2] | 2 [2,3] | 38 [36,39] |
| | FEdit | 58 [54,62] | 2 [1,3] | 22 [20,23] | 3 [2,3] | 62 [60,65] | 99 [98,100] | **34** [30,37] | 40 [37,43] | 2 [2,3] | 5 [3,7] | 33 [32,33] |
| | IFQL | 36 [32,39] | 1 [0,2] | **86** [85,87] | **24** [21,27] | 84 [83,85] | **100** [100,100] | 0 [0,0] | 11 [10,12] | 0 [0,0] | 2 [1,3] | 34 [34,35] |
| ADJOINT MATCHING | QAM | 81 [78,84] | 18 [14,22] | 67 [64,69] | 11 [9,14] | 97 [96,98] | 100 [99,100] | 0 [0,0] | 64 [62,66] | 3 [3,4] | 3 [2,4] | **44** [44,45] |
| | QAM-F | 83 [81,84] | 12 [8,16] | 65 [62,68] | 12 [9,14] | 95 [92,97] | 99 [89,100] | 6 [5,8] | 65 [63,67] | 3 [2,3] | 14 [11,17] | **45** [45,46] |
| | QAM-E | 83 [80,86] | 1 [0,3] | 59 [54,63] | 2 [1,3] | 97 [96,98] | **100** [100,100] | **39** [35,43] | 65 [63,68] | 5 [4,6] | 6 [4,9] | **46** [45,47] |

Table 1: **Offline RL performance at 1M training steps (50 tasks, 12 seeds).** Our method (QAM) and two of its variants, QAM-FQL (QAM-F) and QAM-EDIT (QAM-E) outperform all prior baselines. We use CGQL-M, CGQL-L, QAM-F, QAM-E as the abbreviations for CGQL-MSE, CGQL-Linex, QAM-FQL, and QAM-EDIT respectively. For results on individual tasks, see Table 2 in Section B.

which enjoys the guarantee that the policy at the optimum recovers the optimal behavior-constrained policy, whereas FBRAC has not been found to enjoy such guarantee. This suggests that carefully designing the loss function where its optimum coincides with the optimal behavior-constrained policy can be influential to the algorithm's performance. Furthermore, BAM ends up performing similarly to FQL, which addresses the BPTT instability issue by leveraging a 1-step distilled policy. This suggests that BPTT might not be as disadvantageous as previously reported in Park et al. (2025c) in the best case, but can be sensitive to implementation details. In addition, we find QSM, one of the earliest methods proposed in the literature of diffusion/flow-matching and RL can perform very well when augmented with a behavior cloning term. [2] We also find that a Gaussian baseline, ReBRAC, while worse on manipulation tasks, often outperforms other flow/diffusion-based methods on locomotion tasks. This makes ReBRAC the fifth-best method in terms of aggregated score, falling short compared to QSM and all three QAM variants. At last but not least, combining QAM with FQL and FEdit (QAM-F and QAM-E respectively) can push the performance even further, achieving an aggregated score of 45 and 46 respectively.

**(Q2) How effective is our method for offline-to-online fine-tuning?**

Next, we take the best performing variant of QAM, QAM-EDIT, and evaluate its ability to online fine-tune from its offline RL initialization. Figure 2 shows the sample efficiency curve (with x-axis being the number of environment steps). Our method outperforms all prior methods on cube-triple and is the most robust method across the board. For example, compared to ours, QSM fine-tunes better on ag but struggles on all other tasks except on c2 where it performs similar to our method. FQL fine-tunes slightly better on ag but much slower on both p44 and c3.

**(Q3) How sensitive is our method to hyperparameters?** We also conduct sensitivity analyses for various components of our best performing variant, QAM-EDIT, including gradient clipping, number of flow steps ($T$), critic ensemble size ($K$) and the temperature coefficient ($\tau$). As we show

---

[2]The original QSM implementation does not have a behavior cloning term. It only aligns the action gradient of the critic to the diffusion denoising process. This makes the vanilla QSM very hard to learn anything during the offline pre-training phase. To adapt it to our offline-to-onlilne setting and make the comparison fair, we augment the original QSM loss with the standard DDPM loss on as the behavior cloning loss. See more details in Section E.

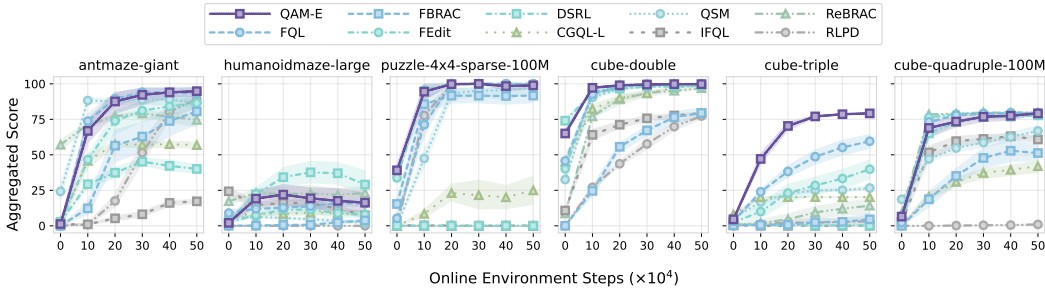

Figure 2: **QAM online fine-tunes more effectively than prior methods (50 tasks, 12 seeds).** We use the `QAM-EDIT` (QAM-E) variant for `QAM` and the `CGQL-Linex` variant for CGQL (CGQL-L) because they work well for online fine-tuning. For full results, see Figure 5 in the Appendix.

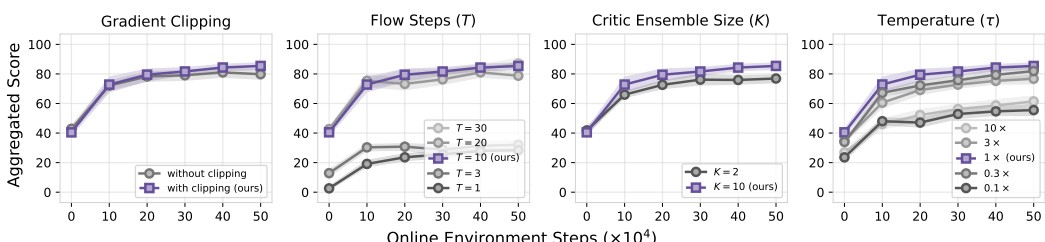

Figure 3: **Sensitivity analysis of `QAM-EDIT` on the representative task from each domain (12 seeds).** For all locomotions tasks, we use `task1`. For all manipulation tasks, we use `task2`. We report the aggregated performance across these 10 representative tasks. *Gradient Clipping*: whether to use gradient clipping in our optimizer; *Flow Steps ($T$)*: this parameter indicates the number of numerical integration steps that we use for the flow model. *Critic Ensemble Size ($K$)*: number of critic network in the ensemble; *Temperature ($\tau$)*: the parameter that modulates the influence of the prior. We rerun our method with $0.1\times, 0.3\times, 3\times$, and $10\times$ the best $\tau$ from tuning. Full results in Figure 6.

in Figure 3, all of these components contribute to our method's effectiveness. Among them, the temperature parameter ($\tau$) has the biggest impact on performance and need to be tuned. For the other components, we find enabling gradient clipping and having a large critic ensemble size ($K = 10$) also help. Lastly, we find that setting the number of flow steps to $T = 10$ works well enough and increasing it further does not improve performance.

# 7 DISCUSSION

We present Q-learning with Adjoint Matching (QAM), a new TD-based RL method that effectively leverages the critic's action gradient to extract an optimal prior-constrained policy while circumventing common limitations of prior approaches (*e.g.*, approximations that do not guarantee to converge to the desired optimal solution, learning instability, or reduced expressivity from distillation). Our empirical results suggest that QAM is an effective policy extraction method in both the offline RL setting and the offline-to-online RL setting, performing on par or better than prior methods. There are still practical challenges associated with QAM. While QAM's effectiveness can be largely attributed to how well it is able to leverage the critic's action gradient, this can be a double-edge sword—for cases where the critic function is ill-conditioned, it could lead to optimization stability issue. Gradient clipping (as done in our method) can alleviate this issue, but a more principled method that combines both value and gradient information could further improve robustness and performance. Another possible extension is to apply QAM in real-world robotic settings with action chunking policies. Our initial success (especially in the manipulation domains where we also leverage action chunking policies) may suggest that our method might work more effectively in complex real-world scenarios.

ACKNOWLEDGMENTS

This research was partly supported by DARPA ANSR and ONR N00014-25-1-2060. This research used the Savio computational cluster resource provided by the Berkeley Research Computing program at UC Berkeley. We would like to thank William Chen, Kevin Frans, David Ma, Vivek Myers, Seohong Park, Michael Psenka, Ameesh Shah, Max Wilcoxson, Charles Xu, Bill Zheng, Zhiyuan (Paul) Zhou, Lars Ankile for helpful discussion and feedback on early drafts of the paper and Catherine Glossop for spotting the bug in an early version of the QSM implementation. We would also like to thank anonymous ICLR reviewers for the insightful comments and discussions.

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

## A  ADDITIONAL RESULTS

**How robust is our method to data corruption?**  We study how robust `QAM-EDIT` is to data corruptions. In particular, we take the `noisy` (for manipulation tasks) and `stitch` (for navigation tasks) datasets from OGBench. The `noisy`-style datasets were collected by expert policies with larger, uncorrelated Gaussian noise, resulting in broader state coverage but larger learning challenge. The `stitch`-style datasets were collected by the same expert policies as the original `navigation`-style datasets (what we use in all our other experiments), but the trajectories are much shorter segments (traveling up to 4 cells in the maze). These variants stress test the algorithm's ability to learn from noisy trajectories and 'stitch' short trajectory segments. We evaluate our method and compare it against a representative set of best-performing baselines. We use the same hyperparameters from our main experiments for each domain and report the results in Figure 4. For locomotion tasks, we observe almost no performance difference when switching from `navigate`-style to `stitch`-style dataset. For manipulation tasks, our method exhibits strong robustness against action noises. In particular, while all our baselines (except BAM) fail completely on the `cube-triple-noisy` environments, our method only suffers minor performance drop.

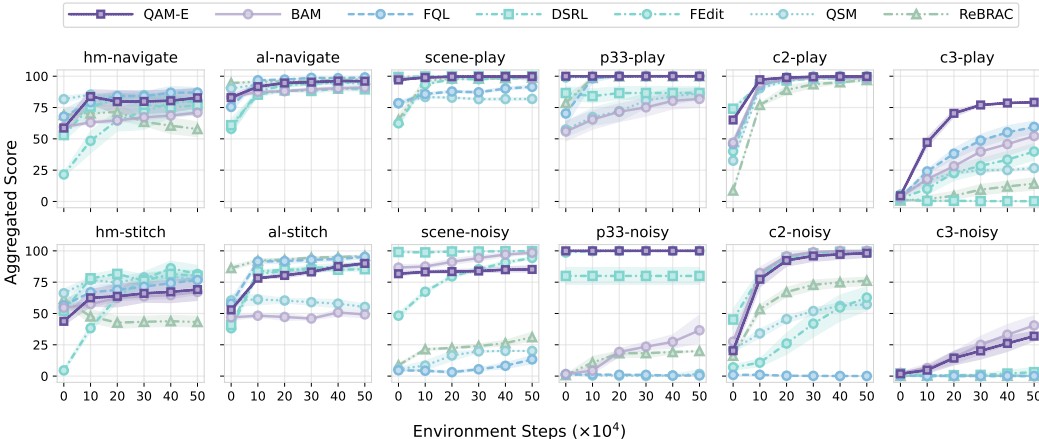

Figure 4: **Data quality analysis (12 seeds).** *Top:* performance on the original {`navigate/play`}-style datasets we use in our main experiments; *Bottom:* performance on the {`stitch/noisy`}-style datasets. Each subplot reports the aggregated score over 5 tasks in each domain. For results on individual tasks and full training curves offline, see Figure 7 below.

## B  FULL RESULTS

We provide full training curves and result table for all methods and tasks individually. Figure 5 shows both the offline phase (until 1M steps) and the online phase (after 1M steps). Table 2 reports the performance at the end of the 1M-step offline training. The hyperparameters can be found in Section F. Figure 6 shows the full training curves for our sensitivity analysis.

## C  ADDITIONAL DISCUSSIONS FOR RELATED WORK

**Diffusion and flow-matching with guidance.** Diffusion and flow-matching models have been used for generating data with different modalities ranging from images (Rombach et al., 2022), videos (Ho et al., 2022), and text (Lou et al., 2024). In most applications, the generative models trained on large-scale unlabeled data do not provide high-quality samples when conditioned on some context (*e.g.*, language description), and a common practice is to augment the sampling process with classifier guidance/classifier-free guidance (Dhariwal & Nichol, 2021; Ho & Salimans, 2022), with the goal of aligning the sampling distribution better with the posterior distribution conditioned on the context. However, most of the guidance methods suffer from a bias problem that is tricky to tackle, stemming from the fact that simply adding or interpolating two diffusion/flow-matching sampling processes

does not lead to the correct composite distribution in general (Du et al., 2023; Bradley & Nakkiran, 2024). One solution is to use Langevin dynamics sampling approaches (Song & Ermon, 2019) where only the score function for the noise-free distribution is required, but they have been known to under-perform diffusion/flow models due to the challenge of accurately estimating the score functions in low-density regions (Song & Ermon, 2020). Since then, a line of work has proposed solutions to generate the correct composite distribution. Du et al. (2023), Phillips et al. (2024), Thornton et al. (2025), Singhal et al. (2025) and Skreta et al. (2025) propose to use Sequential Monte Carlo (SMC) that uses resampling procedures to leverage additional test-time compute to correct such bias. Rather than correcting the distribution at test-time, Domingo-Enrich et al. (2025) and Havens et al. (2025) take a different perspective by formulating it as a stochastic optimal control (SOC) objective that can be efficiently optimized as a fine-tuning process while providing guarantee that the model converges to the correct distribution at the optimum. The flow/diffusion policy optimization problem in actor-critic RL methods shares a similarity to the aforementioned classifier/classifier-free guidance problem in generative modeling literature where the critic function serves as the guidance to the generative policy model. This allows our method builds directly on top of the algorithm developed by Domingo-Enrich et al. (2025) while enjoying the guarantee that the policy converges to an optimal prior-regularized solution (*i.e.*, $\pi \propto \pi_\beta \exp(Q(s, a))$).

**CEP (Lu et al., 2023a) and CFGRL (Frans et al., 2025).** Both of them build off from the idea of classifier/classifier-free guidance, which combines the denoising step of a base diffusion/flow policy to the denoising step of for a tilt distribution.

$$\text{CEP (diffusion):} \quad \log \pi(s, a_t, t) \leftarrow \alpha \log \pi_\beta(s, a_t, t) + (1 - \alpha) Q(s, a_t, t) \tag{32}$$

$$\text{CFGRL (flow):} \quad f(s, a_t, t) \leftarrow \alpha f_\beta(s, a_t, t) + (1 - \alpha) f_{o=1}(s, a_t, t) \tag{33}$$

where $Q(s, a_t, t) \leftarrow \log \mathbb{E}_{a_t|a} \left[ e^{Q(s,a)} \right]$ is the score of the Boltzmann distribution (i.e., $\propto e^{Q(s,a)}$) at denoising time $t$ and $f_o$ is the velocity field of the policy that is conditioned on a optimality variable, a binary indicator of whether the policy is 'optimal' ($o = 1$ means it is). CEP aims at approximating $\pi \propto (\pi_\beta)^\alpha (e^{Q(s,a)})^{(1-\alpha)}$ whereas CFGRL aims at approximating $\pi \propto (\pi_\beta)^\alpha (\pi_{o=1})^{(1-\alpha)}$.

However, as discussed in many prior work (Du et al., 2023; Bradley & Nakkiran, 2024), even when both the denoising steps are exact ($\log \pi(s, a_t, t)$ for diffusion and $f(\cdot, \cdot, t)$ for flow), the denoising process that uses a summation of them do not lead to the correct distribution:

$$\nabla_{a_t} \log \pi(a_t \mid s) \neq \nabla_{a_t} \log \pi_\beta(a_t \mid s) + \tau \nabla_{a_t} Q(s, a_t). \tag{34}$$

**DAC (Fang et al., 2025b).** Diffusion actor critic uses the diffusion formulation where the goal is to find a policy that satisfies $\pi(\cdot \mid s) \propto \pi_\beta(\cdot \mid s) e^{Q(s,\cdot)}$. However, their training objective is derived based on the assumption that

$$\nabla_{a_t} \log p_t(a_t \mid s) \approx \nabla_{a_t} \log p_\beta(a_t \mid s) + \tau \nabla_{a_t} Q_t(s, a_t), \tag{35}$$

and additionally

$$\nabla_{a_t} Q_t(s, a_t) \approx \nabla_{a_t} Q(s, a_t). \tag{36}$$

While these assumptions provide a convenient approximation of the objective function, it does not provide guarantees on where policy converges to at the optimum.

## D   DOMAIN AND EXPERIMENT DETAILS

We consider 10 domains in our experiments. The dataset size, episode length, and the action dimension for each domain is available in Table 3. For each method and each task, we run 12 seeds. All plots and tables report the means with $95\%$ confidence intervals computed via bootstrapping using 5000 samples. All our experiments are run on NVIDIA RTX-A5000 GPU and our code is written in JAX (Bradbury et al., 2018). We use NVIDIA-A5000 GPU to run all our experiments. Each complete offline-to-online experiment run takes around 3 hours. To reproduce all our results in Table 1 and Figure 2, we estimate that it would take around $\underbrace{5}_{\text{hours per single run}} \times \underbrace{17}_{\text{\# of methods}} \times \underbrace{50}_{\text{\# of tasks}} \times \underbrace{12}_{\text{\# of seeds}} = 51\,000$ GPU hours.

# E  BASELINES

In this section, we describe in detail how each of our baselines are implemented. In all the loss functions below, unless specified otherwise, $s, a, r, s'$ are assumed to be sampled from $D$ and as part of the expectation even when it is not explicitly written under the expectation (*i.e.*, $\mathbb{E}_{z \sim \mathcal{N}}[\cdot] := \mathbb{E}_{(s,a,r,s') \sim D, z \sim \mathcal{N}}[\cdot]$).

**1. Backprop-based.**

**FBRAC** is a baseline considered in FQL (Park et al., 2025c) as a flow counterpart of diffusion Q-learning (DQL) (Wang et al., 2023), where the multi-step flow policy is directly optimized against the Q-function with backpropagation through time (BPTT). In addition to maximizing the Q-value, FBRAC also has a behavior cloning term where the flow policy is trained with the standard flow-matching objective on the dataset action with a BC coefficient $\alpha$.

We implement this baseline by training a flow-matching policy with the following loss:

$$L_{\texttt{FBRAC}}(\theta) = \alpha L_{\mathrm{FM}}(\theta) + L_{\mathrm{BPTT}}(\theta), \tag{37}$$

where $L_{\mathrm{FM}}$ is the standard flow-matching loss that clones the data behavior (*i.e.*, Equation (17)) and

$$L_{\mathrm{BPTT}}(\theta) = -\mathbb{E}_{a^0 = z \sim \mathcal{N}(0, I_A)} \left[ Q_\phi \left( s, \mathrm{Clip} \left[ z + h \sum_{k=0}^{T-1} f_\theta(s, a^{k-1}, kh) \right]_{-1}^{1} \right) \right], \tag{38}$$

where $\mathrm{Clip}[\cdot]_a^b$ is an element-wise clipping function that makes sure the actions generated from the flow model $f_\theta$ are within the valid range $[-1, 1]$ and $\{a^i\}_i$ is the discrete approximation of the ODE trajectory using Euler's method with a step size of $h = 1/T$:

$$a^{i+1} := a^i + h f_\theta(s, a^{i-1}, ih), \forall i \in \{0, 1, \cdots, T\}. \tag{39}$$

We use

$$\mathrm{ODE}(f_\theta(s, \cdot, \cdot), z) := \mathrm{Clip} \left[ z + h \sum_{i=0}^{T-1} f_\theta(s, a^{i-1}, ih) \right]_{-1}^{1} \tag{40}$$

as the abbreviation for the rest of this section, and additionally use $\pi_{f_\theta}$ to denote the distribution of actions generated by $f_\theta$.

The critic loss is the standard TD backup:

$$L(\phi) = \mathbb{E}_{z \sim \mathcal{N}} \left[ (Q_\phi(s, a) - r - Q_{\bar\phi}(s', \mathrm{ODE}(f_\theta(s', \cdot, \cdot), z)))^2 \right] \tag{41}$$

In practice, we also use $K = 10$ critic functions and pessimistic target value backup as described in Equation (26) and the policy we use to interact with the environment is $\pi_{f_\theta}$.

**FQL** (Park et al., 2025c) distill a multi-step flow policy into a one-step distillation to avoid BPTT.

This baseline is implemented by training a behavior cloning flow-matching policy (*i.e.*, $f_\theta : \mathcal{S} \times \mathbb{R}^A \times [0, 1] \to \mathbb{R}^A$), and a 1-step distilled noise-conditioned policy (*i.e.*, $\Omega_\omega : \mathcal{S} \times \mathbb{R}^A \to \mathbb{R}^A$):

$$L_{\texttt{FQL}}(\theta, \omega) = L_{\mathrm{FM}}(\theta) + L_{\mathrm{onestep}}(\omega) \tag{42}$$

where $L_{\mathrm{FM}}(\theta)$ is the standard flow-matching loss (i.e., Equation (17)) and

$$L_{\mathrm{onestep}}(\omega) = \mathbb{E}_{z \sim \mathcal{Z}} \left[ \alpha \| \Omega_\omega(s, z) - \mathrm{ODE}(f_\theta(s', \cdot, \cdot), z) \|_2^2 - Q(s, \Omega_\omega(s, z)) \right] \tag{43}$$

where $\alpha$ is the BC coefficient that controls how close the 1-step distilled policy should be relative to the BC policy $\pi_{f_\theta}$. Finally, the critic loss is the standard TD backup:

$$L(\phi) = \mathbb{E}_{z \sim \mathcal{N}} \left[ (Q_\phi(s, a) - r - Q_{\bar\phi}(s', \Omega_\omega(s', z)))^2 \right] \tag{44}$$

In practice, we also use $K = 10$ critic functions and pessimistic target value backup as described in Equation (26) and the policy we use to interact with the environment is $\pi_{\Omega_\omega}$ where the action is sampled by first drawing a Gaussian noise $z \sim \mathcal{N}$ and then obtain the action by running through the one-step distilled model: $a = \Omega_\omega(s, z)$.

**2. Directly using the action gradient of the critic (i.e., $\nabla_a Q(s,a)$) with approximations.**

**QSM** (Psenka et al., 2024b) uses the action gradient of the critic to train a diffusion model such that it can reconstruct the policy that follows the Boltzmann distribution of the Q-function:

$$\pi(\cdot \mid s) \propto \exp(\tau Q_\phi(s, \cdot)). \tag{45}$$

To approximate the score of the intermediate actions, $\nabla_{a^i} \log \pi^i(a^i \mid s)$, QSM directly uses the critic evaluated at $a^i$:

$$\nabla_{a^i} \log \pi^i(a^i \mid s) \approx \tau \nabla_{a^i} Q_\phi(s, a^i) \tag{46}$$

where $a^i = \sqrt{\alpha^i} a + \sqrt{1 - \alpha^i} \varepsilon$ with $\varepsilon \sim \mathcal{N}$ and $\{\alpha^0, \cdots, \alpha^{T-1}\}$ being any diffusion schedule with the number of diffusion steps of $T$. For all diffusion methods in this paper, we follow Hansen-Estruch et al. (2023) to use the variance preserving diffusion schedule:

$$\beta^i = 1 - \exp\left(-\frac{b_{\min}}{T} - \frac{(b_{\max} - b_{\min})(2i + 1)}{2T^2}\right), \tag{47}$$

$$\alpha^i = \prod_{j=0}^{i}(1 - \beta^j), \tag{48}$$

for all $i \in \{0, 1, \cdots, T - 1\}$ with $b_{\min} = 0.1, b_{\max} = 10$ and $T$ being the number of diffusion steps.

To train the diffusion model, QSM uses the following loss function:

$$L_{\text{QSM}}(\theta) = \mathbb{E}_{\varepsilon \sim \mathcal{N}, i \sim \mathcal{U}\{0, \cdots, T-1\}} \left[\| - \tau \nabla_a Q_\phi(s, a^i) - f_\theta(s, a^i, i)\|_2^2\right], \tag{49}$$

where again $a^i = \sqrt{\alpha^i} a + \sqrt{1 - \alpha^i} \varepsilon$. In practice, we find that using $\nabla_a Q_{\bar{\phi}}(s, a^i)$ (the action gradient of the critic network) helps learning stability, so we use that instead.

To adopt QSM into the offline-to-online RL setting, we additionally augment the loss function with the standard diffusion loss (Ho et al., 2020) as the behavior regularization:

$$L_{\text{DDPM}}(\theta) = \mathbb{E}_{\varepsilon \sim \mathcal{N}, i \sim \mathcal{U}\{0, \cdots, T-1\}} \left[\|\varepsilon - f_\theta(s, a^i, i)\|_2^2\right]. \tag{50}$$

The overall actor loss is thus

$$L(\theta) = L_{\text{QSM}}(\theta) + \eta L_{\text{DDPM}}(\theta). \tag{51}$$

QSM then uses the standard diffusion denoising procedure (with the clipping to make sure generated actions are within $[-1, 1]^{|A|}$):

$$a^{i-1} \leftarrow \text{Clip}\left[\frac{1}{\sqrt{1 - \beta^i}}\left(a^i - \frac{\beta^i}{\sqrt{1 - \alpha^i}} f_\theta(s, a^i, i)\right) + \sqrt{\beta^i}\varepsilon^i\right]_{-1}^{1}, \quad \varepsilon^i \sim \mathcal{N}, \tag{52}$$

where $a^T \sim \mathcal{N}$. We denote $\text{Diff}(f_\theta(s, \cdot, \cdot))$ as the resulting $a^0$ after going through the diffusion procedure above and $\pi_{f_\theta} : s \mapsto \text{Diff}(f_\theta(s, \cdot, \cdot))$ as the policy that generates actions using $f_\theta$.

In practice, we find that the following a modified diffusion denoising procedure yields better performance. In particular, we use the $\hat{a}^{i-1}$ from DAC,

$$\hat{a}^{i-1} \leftarrow \text{Clip}\left[\frac{1}{\sqrt{\alpha^i}}\left(a^i - \sqrt{1 - \alpha^i} f_\theta(s, a^i, i)\right)\right]_{-1}^{1}, \tag{53}$$

and directly set $a^{i-1}$ as

$$a^{i-1} \leftarrow \text{Clip}\left[\hat{a}^{i-1} + \sqrt{\beta^i}\varepsilon^i\right]_{-1}^{1} \tag{54}$$

for the next diffusion iteration. Even though this deviates from the standard diffusion denoising procedure, and may seem arbitrary, we empirically find that this modified version to produce much better offline RL performance. One hypothesis is that directly using $\hat{a}^{i-1}$ allows us to evaluate action gradient (*i.e.*, $\nabla_a Q(s, a)$) on actions that are more likely to be in-distribution.

The critic loss can now be defined as follows:

$$L(\phi) = \mathbb{E}_{z \sim \mathcal{N}} \left[ (Q_\phi(s,a) - r - Q_{\bar{\phi}}(s', a' \sim \pi_{f_\theta}(\cdot \mid s')))^2 \right] \tag{55}$$

In practice, we also use $K = 10$ critic functions and pessimistic target value backup as described in Equation (26) and the policy we use to interact with the environment is $\pi_{f_\theta}$.

**DAC** (Fang et al., 2025b) is another method that uses a similar approximation with a behavior prior (*i.e.*, $\nabla_{a^i} \log \pi^i(a^i \mid s) \approx \nabla_{a^i} \log \pi_\beta^i(a^i \mid s) + \tau \nabla_{a^i} Q_\phi(s, a^i)$). To train the diffusion model, DAC matches $f_\theta$ to a linear combination of $\varepsilon$ (behavior cloning) and $\nabla_{a^i} Q(s, a^i)$ (Q-maximization).

$$\begin{aligned} L_{\mathtt{DAC}}(\theta) &= \mathbb{E}_{\varepsilon \sim \mathcal{N}, i \sim \mathcal{U}\{0, \cdots, T-1\}} \left[ \| \varepsilon - \tau \nabla_{a^i} Q_\phi(s, a^i) - f_\theta(s, a^i, i) \|_2^2 \right] \\ &= \mathbb{E}_{\varepsilon \sim \mathcal{N}, i \sim \mathcal{U}\{0, \cdots, T-1\}} \left[ \| \varepsilon - f_\theta(s, a^i, i) \|_2^2 + \eta^{-1}(f_\theta(s, a^i, i) \cdot \nabla_{a^i} Q_\phi(s, a^i)) \right] + C, \end{aligned} \tag{56}$$

with $\eta = 2/\tau$. In practice, it is implemented as a combination of

$$L(\theta) = \mathbb{E}_{\varepsilon \sim \mathcal{N}, i \sim \mathcal{U}\{0, \cdots, T-1\}} \left[ \| (f_\theta(s, a^i, i) \cdot \nabla_{a^i} Q_\phi(s, a^i)) \right] + \eta L_{\mathrm{DDPM}}(\theta) \tag{57}$$

DAC uses the following procedure:

$$\hat{a}^{i-1} \leftarrow \mathrm{Clip} \left[ \frac{1}{\sqrt{\alpha^i}} \left( a^i - \sqrt{1 - \alpha^i} f_\theta(s, a^i, i) \right) \right]_{-1}^{1} \tag{58}$$

$$a^{i-1} \leftarrow \frac{\beta^i \sqrt{\alpha^{i-1}}}{1 - \alpha^i} \hat{a}^{i-1} + \sqrt{1 - \beta^i}(1 - \alpha^{i-1}) a^i + \sqrt{\beta^i} \varepsilon^i, \quad \varepsilon^i \sim \mathcal{N} \tag{59}$$

where $a^T \sim \mathcal{N}$ and $\hat{a}^{i-1}$ can be interpreted as an approximation of the denoised action (*e.g.*, in the DDIM sampler (Song et al., 2020)). This approximated denoised action is first clipped to be within $[-1, 1]^{|A|}$ and then used to reconstruct $a^{i-1}$ using the closed-form Gaussian conditional probability distribution of $p(a^{i-1} \mid a^0 = \hat{a}^{i-1}, a^T = a)$ (*e.g.*, Eq. (7) in Ho et al. (2020)). In practice, DAC also uses $\sqrt{\beta^i}$ as standard deviation for this conditional distribution instead of the correct one $\tilde{\beta}^i = \frac{\sqrt{1-\alpha^{i-1}}}{\sqrt{1-\alpha^i}} \beta^i$ as an approximation. This is why in the expression above the multiplier before $\varepsilon^i$ is $\sqrt{\beta^i}$.

The critic loss can now be defined as follows:

$$L(\phi) = \mathbb{E}_{z \sim \mathcal{N}} \left[ (Q_\phi(s,a) - r - Q_{\bar{\phi}}(s', a' \sim \pi_{f_\theta}(\cdot \mid s')))^2 \right], \tag{60}$$

where $\pi_{f_\theta}$ is the policy that generates actions using the procedure above in Equation (58) and Equation (59). In practice, we also use $K = 10$ critic functions and pessimistic target value backup as described in Equation (26) and the policy we use to interact with the environment is $\pi_{f_\theta}$.

**CGQL** is a novel baseline built on top of the idea of classifier guidance (Dhariwal & Nichol, 2021). In particular, we combine the velocity field of a behavior cloning flow policy and the gradient field of the Q-function to form a new velocity field that approximates the velocity field that generates the optimal behavior-constrained action distribution.

More specifically, we implement this baseline by interpreting $Q_\phi(s, \cdot)$ as the score of the optimal entropy-regularized distribution $\log \pi^\star(\cdot \mid s)$ (where $\pi^\star(\cdot \mid s) \propto e^{\tau Q_\phi(s, \cdot)}$). The corresponding velocity field that generates this distribution of actions can be obtained through a simple conversion (*e.g.*, following Equation 4.79 from Lipman et al. (2024)):

$$v_\phi(s, a, t) := \frac{(1-t)\tau \nabla_a Q(s, a, t) + a}{t}, \tag{61}$$

where

$$Q(s, a_t, t) := \frac{1}{\tau} \log \mathbb{E}_{a_1 \sim P(a_1 \mid a_t)} \left[ e^{\tau Q_\phi(s, a_1)} \right], \tag{62}$$

is an approximation of the score of the distribution over the noisy intermediate actions. In the classifier guidance literature, the score of the noisy examples is approximated by the score of the noise-free examples at the noisy examples (Dhariwal & Nichol, 2021). In our setting this translates to

$$\hat{f}_\phi(s, a_t, t) := \frac{(1-t)\tau \nabla_a Q_\phi(s, a_t) + a_t}{t}. \tag{63}$$

Empirically, both versions ($f_\phi$ and $\hat{f}_\phi$) perform similarly and we opt for a simpler design $\hat{f}$ as it does not require learning or approximating $Q(s, a, t)$ for all $u \in [0, 1]$. Finally, we add the velocity field defined by $Q_\phi$ directly to the behavior cloning velocity field to form our policy:

$$f = f_\beta + \vartheta \hat{f}_\phi, \tag{64}$$

where $f_\beta$ is trained with the standard flow-matching loss (i.e., $L(\beta) = L_{\mathrm{FM}}(\beta)$) and $\vartheta$ is coefficient that modulates influence of the guidance that we find to be helpful (e.g., $\vartheta < 1$ often works better than $\vartheta = 1$). The critic loss uses $\pi_v$ to backup the target $Q$-value:

$$L(\phi) = \mathbb{E}_{z \sim \mathcal{N}} \left[ (Q_\phi(s, a) - r - Q_{\bar{\phi}}(s', \mathrm{ODE}(f(s', \cdot, \cdot), z)))^2 \right]. \tag{65}$$

In practice, we also use $K = 10$ critic functions and pessimistic target value backup as described in Equation (26) and the policy we use to interact with the environment is $\pi_v$ (generated from the summation of $f_\beta$ and $\hat{f}_\phi$).

**CGQL-MSE/Linex.** Alternatively, we can approximate $Q_t$ in Equation (62) more closely with a training objective. In particular, we explore the following two regression objectives:

$$\textbf{MSE:} \quad L_{\mathrm{MSE}}(\zeta) = \mathbb{E}_{t \sim \mathcal{U}[0,1], z \sim \mathcal{N}} \left[ \left( \hat{Q}_\zeta(s, a_t, t) - Q_\phi(s, a) \right)^2 \right] \tag{66}$$

$$\textbf{Linex:} \quad L_{\mathrm{Linex}}(\zeta) = \mathbb{E}_{t \sim \mathcal{U}[0,1], z \sim \mathcal{N}} \left[ \exp(\tau(Q_\phi(s, a) - \hat{Q}_\xi(s, a_t, t))) + \tau \hat{Q}_\xi(s, a_t, t) \right] \tag{67}$$

where $a_t := (1 - t)z + ta$. The optimal solution of $\hat{Q}_\zeta$ in Equation (66) is not exactly the same as the desired $Q_\phi$ in Equation (62) but constitutes a lower-bound due to Jensen's inequality:

$$Q_{\mathrm{MSE}}^\star(s, a_t, t) = \mathbb{E}_{z,t} \left[ Q_\phi(s, a_t) \right] \le \frac{1}{\tau} \log \mathbb{E}_{a_1 \sim P(a_1 | a_t)} \left[ e^{\tau Q_\phi(s, a_t)} \right]. \tag{68}$$

The second objective resembles the classic Linex objective (Parsian & Kirmani, 2002) where the optimal solution of $\hat{Q}_\zeta$ in Equation (67) is the same as the desired $Q$ in Equation (62):

$$Q_{\mathrm{Linex}}^\star(s, a_t, t) = \frac{1}{\tau} \log \mathbb{E}_{a_1 \sim P(a_1 | a_t)} \left[ e^{\tau Q_\phi(s, a_1)} \right]. \tag{69}$$

A discussion on this can be found in Myers et al. (2025). For completeness, we also show this below. Without loss of generality, we just need to show that $\exp(y) = \mathbb{E}[\exp(x)] = \int p(x) \exp(x) \mathrm{d}x$ is the unique minimum for the following loss function:

$$L(y) = \int p(x) \left[ \exp(x - y) + y \right] \mathrm{d}x \tag{70}$$

Taking the first and second derivatives with respect to $y$ gives

$$\frac{\mathrm{d}L}{\mathrm{d}y} = -\exp(-y)\mathbb{E}[\exp(x)] + 1 \tag{71}$$

$$\frac{\mathrm{d}^2 L}{\mathrm{d}y^2} = \exp(-y)\mathbb{E}[\exp(x)] \tag{72}$$

Setting Equation (71) gives $y = \log \mathbb{E}[\exp(x)]$ at which the second derivative is 1 (i.e., $\frac{\mathrm{d}^2 L}{\mathrm{d}y^2} = 1$). Furthermore, to prevent the exponential blow up of $\exp(Q_\phi(s, a) - \hat{Q}_\xi(s, a_t, t))$, we follow Myers et al. (2025) to use a Huber-style loss that locally behaves like a Linex loss but with a linear penalty when the exponential term is too large. In particular, we use the following loss:

$$L_{\mathrm{Linex}^+}(\zeta) = \begin{cases} \mathbb{E} \left[ \exp(\Delta) + \tau \hat{Q}_\xi(s, a_t, t) \right], & \Delta < 5 \\ \mathbb{E} \left[ \Delta \right], & \Delta \ge 5 \end{cases} \tag{73}$$

where $\Delta = \tau(Q_\phi(s, a) - \hat{Q}_\xi(s, a_t, t))$.

In practice, however, both the MSE and Linex objectives ($L_{\mathrm{Linex}^+}, L_{\mathrm{MSE}^+}$) can still be unstable and exhibit high variances. Instead of learning $\hat{Q}_\zeta$ and $Q_\phi$ separately, we find it is often better to directly learn both of them in a single network $Q_\phi(s, a, t) : \mathcal{S} \times \mathcal{A} \times [0, 1] \to \mathbb{R}$ as follows:

$$\begin{aligned} L(\phi) = &\mathbb{E}_{z \sim \mathcal{N}} \left[ (Q_\phi(s, a, 1) - r - Q_{\bar{\phi}}(s', \mathrm{ODE}(f(s', \cdot, \cdot), z)))^2 \right] + \\ &\varrho \mathbb{E}_{t \in \mathcal{U}[0,1], z \sim \mathcal{N}} \left[ (Q_\phi(s, (1 - t)z + ta, t) - Q_{\bar{\phi}}(s, a, 1))^2 \right], \end{aligned} \tag{74}$$

where $\varrho$ is the coefficient that balances the TD backup for $Q_\phi(s, a, 1)$ and the noisy target regression for $Q_\phi(s, a, t < 1)$. With $Q_\phi(s, \cdot, t)$, we can define our velocity field (which is also used as the TD backup above) as

$$f := f_\beta + \vartheta \hat{f}_\phi, \quad \text{where } \hat{f}_\phi(s, a_t, t) := \frac{\tau \nabla_a Q_\phi(s, a_t, t) + a_t}{t}, \tag{75}$$

and $\vartheta$ again modulates the guidance strength.

**3. Directly using the critic value (i.e., $Q(s, a)$).**

**FAWAC** is a baseline considered in FQL (Park et al., 2025c) where it uses AWR to train the flow policy similar to QIPO (Zhang et al., 2025).

We implement it by training a flow-matching policy with the weighted flow-matching loss:

$$L_{\texttt{FAWAC}}(\theta) = \tilde{w}(s, a) L_{\text{FM}}(\theta) \tag{76}$$

$$= \tilde{w}(s, a) \mathbb{E}_{t \sim \mathcal{U}[0,1], z \sim \mathcal{N}} \left[ \| f_\theta(s, (1-t)z + ta, t) - z + a \|_2^2 \right] \tag{77}$$

where $\tilde{w}(s, a) = \min \left( e^{\tau(Q_\phi(s,a) - V_\xi(s))}, 100.0 \right)$. The inverse temperature parameter $\tau$ controls how sharp the prior regularized optimal policy distribution is.

The critic function $Q_\phi(s, a)$ is trained with the standard TD backup and the value function $V_\xi(s)$ regresses to the same target:

$$L(\phi) = \mathbb{E}_{z \sim \mathcal{N}} \left[ (Q_\phi(s, a) - r - Q_{\bar{\phi}}(s', \text{ODE}(f(s', \cdot, \cdot), z)))^2 \right] \tag{78}$$

$$L(\xi) = \mathbb{E}_{z \sim \mathcal{N}} \left[ (V_\xi(s) - r - Q_{\bar{\phi}}(s', \text{ODE}(f(s', \cdot, \cdot), z)))^2 \right] \tag{79}$$

The second line can also be alternatively implemented by regressing to the critic function $Q(s, a)$ directly. We implement in this particular way because we can re-use the $Q$-target computed.

In practice, we also use $K = 10$ critic functions and pessimistic target value backup as described in Equation (26) and the policy we use to interact with the environment is $\pi_{f_\theta}$.

**4. Post-processing-based.**

**FEdit** is a baseline that uses the policy edit from a recent offline-to-online RL method conceptually similar to EXPO (Dong et al., 2025). We implement a Gaussian edit policy on top of a BC flow policy rather than a diffusion policy used in EXPO. EXPO also uses the standard sample-and-rank trick where it samples multiple actions and rank them based on the value. To keep computational cost down and comparisons fair to other methods, we only use a single edited action for both value backup and evaluation.

We implement this baseline by training a flow-matching policy (i.e., $f_\theta : \mathcal{S} \times \mathbb{R}^A \times [0, 1] \to \mathbb{R}^A$), and a 1-step Gaussian edit policy (i.e., $\pi_\omega : \mathcal{S} \times \mathcal{A} \to \Delta_\mathcal{A}$) implemented with an entropy regularized SAC policy (Haarnoja et al., 2018). The loss function can be described as follows:

$$L_{\texttt{FEdit}}(\theta, \omega) = L_{\text{FM}}(\theta) + L_{\text{Gaussian}}(\omega), \quad \text{s.t.} \quad \mathbb{E}_{s \sim D} \left[ \mathbb{H}(\pi_\omega(\cdot \mid s)) \right] \geq H_{\text{target}} \tag{80}$$

where $L_{\text{FM}}$ is the standard flow-matching loss that clones the data behavior (i.e., Equation (17)), $H_{\text{target}}$ is the target entropy that the Gaussian policy is constrained to be above of, and

$$L_{\text{Gaussian}}(\omega) = \mathbb{E}_{\Delta a \sim \pi_\omega(\cdot | s, \tilde{a}), z \sim \mathcal{N}} \left[ -Q_\phi(s, \text{Clip} \left[ \sigma_a \cdot \Delta a + \tilde{a} \right]_{-1}^1) \right] \tag{81}$$

where $\tilde{a} = \text{ODE}(f_\theta(s, \cdot, \cdot), z)$ and $\text{Clip}[\cdot]_a^b$ is an element-wise clipping function that makes sure the actions are within the valid range $[-1, 1]$.

Intuitively, the Gaussian SAC policy edits the behavior flow policy by modifying its output action where $\sigma_a$ is the hyperparameter that controls how much the origianl behavior actions can be edited.

The critic loss is the standard TD backup:

$$L(\phi) = \mathbb{E}_{z \sim \mathcal{N}, \Delta a' \sim \pi_\omega(\cdot | s', \tilde{a}')} \left[ (Q_\phi(s, a) - r - Q_{\bar{\phi}}(s', \text{Clip} \left[ \tilde{a}' + \sigma_a \cdot \Delta a' \right]_{-1}^1))^2 \right] \tag{82}$$

where again $\tilde{a}' = \text{ODE}(f_\theta(s', \cdot, \cdot), z)$. In practice, we also use $K = 10$ critic functions and pessimistic target value backup as described in Equation (26) and the policy we use to interact with

the environment is a combination of the BC policy $\pi_{f_\theta}$ and the Gaussian edit policy. We first sample $z \sim \mathcal{N}$ and then run it through the BC flow policy to obtain an initial action $\tilde{a} \leftarrow \text{ODE}(f_\theta(s, \cdot, \cdot), z)$ and then both the initial action and the state is fed into the edit policy to generate the final action $a \leftarrow \tilde{a} + \sigma_a \cdot \Delta a$ where $\Delta a \sim \pi_\omega(\cdot \mid s, \tilde{a})$.

**DSRL** (Wagenmaker et al., 2025) is a recently proposed method that performs RL directly in the noise-space of a pre-trained expressive BC policy (flow or diffusion). We use the flow-matching version of DSRL as our method is also based on flow-matching policies. The original DSRL implementation does not fine-tune the BC policy during online learning while all our baselines do fine-tune the BC policy online. To make the comparison fair, we modify the DSRL implementation such that it also fine-tunes the BC policy. One additional implementation trick that allows this modification to work well is the use of target policy network for the noise-space policy. In general, we find that fine-tuning the BC policy yields better online performance, so we adopt this new design of DSRL in our experiments.

More specifically, we train a flow-matching policy (*i.e.*, $f_\theta : \mathcal{S} \times \mathbb{R}^A \times [0, 1] \to \mathbb{R}^A$), and a 1-step Gaussian edit policy (*i.e.*, $\pi_\omega : \mathcal{S} \times \mathcal{A} \to \Delta_{\mathcal{A}}$) implemented with an entropy regularized SAC policy (Haarnoja et al., 2018). The loss function can be described as follows:

$$L_{\text{DSRL}}(\theta, \omega) = L_{\text{FM}}(\theta) + L_{\text{LatentGaussian}}(\omega), \quad \text{s.t.} \quad \mathbb{E}_{s \sim D}\left[\mathbb{H}(\pi_\omega(\cdot \mid s))\right] \geq H_{\text{target}} \tag{83}$$

where $L_{\text{FM}}$ is the standard flow-matching loss that clones the data behavior (*i.e.*, Equation (17)), $H_{\text{target}}$ is the target entropy that the Gaussian policy is constrained to be above of, and

$$L_{\text{LatentGaussian}}(\omega) = \mathbb{E}_{z \sim \pi_\omega(\cdot \mid s)}\left[-Q_\psi^z(s, z)\right] \tag{84}$$

where $Q_\psi^z(s, z)$ is a distilled critic function in the noise space that is regressed to the original critic function, $Q_\phi(s, a)$:

$$L(\psi) = \mathbb{E}_{z \sim \mathcal{N}}\left[(Q_\psi^z(s, z) - Q_\phi(s, \text{ODE}(f_{\bar\theta}(s, \cdot, \cdot), z))^2\right]. \tag{85}$$

Intuitively, DSRL directly learns a policy in the noise space by hill-climbing a distilled critic that also operates in the noise space. Finally, the critic loss for the original critic function in the action space is

$$L(\phi) = \mathbb{E}_{z \sim \pi_\omega(\cdot \mid s')}\left[(Q_\phi(s, a) - r - Q_{\bar\phi}(s', \text{ODE}(f_{\bar\theta}(s', \cdot, \cdot), z))^2\right] \tag{86}$$

We use $K = 10$ critic functions and pessimistic target value backup as described in Equation (26). The policy we use to interact with the environment is a combination of the BC policy $\pi_{f_\theta}$ and the Gaussian noise-space policy. We first sample $z \sim \pi_\omega(\cdot \mid s)$ and then run it through the BC flow policy to obtain the final action $a \leftarrow \text{ODE}(f_\theta(s, \cdot, \cdot), z)$. One important implementation detail for stability in the offline-to-online setting is to use the target network for the BC flow policy $f_{\bar\theta}$ (instead of $f_\theta$). Without it DSRL can become unstable sometimes when the BC flow policy changes too fast online.

**IFQL** is a baseline considered in FQL (Park et al., 2025c) as a flow counterpart of implicit diffusion Q-learning (IDQL) (Hansen-Estruch et al., 2023), where IQL (Kostrikov et al., 2022) is used for value learning and the policy extraction is done by sampling multiple actions from a behavior cloning diffusion policy and select the one that maximizes the $Q$-function value.

More specifically, we train a critic function $Q_\phi(s, a)$ and a value function $V_\xi(s)$ with implicit value backup:

$$L(\phi) = (Q_\phi(s, a) - r - V_\xi(s'))^2 \tag{87}$$

$$L(\xi) = F_{\exp}^\kappa(Q_{\bar\phi}(s, a) - V_\xi(s)) \tag{88}$$

where $F_{\exp}^\kappa(u) = |\kappa - \mathbb{I}_{u < 0}|u^2$ is the expectile regression loss function.

On top of that, we also use $K = 10$ critic functions and pessimistic target value backup as described in Equation (26) for training the value function $V_\xi(s)$. To extract a policy from $Q_\phi(s, a)$, IFQL uses rejection sampling with a base behavior cloning flow policy that is trained with the standard flow-matching objective. In particular, the output action $a^\star$ for $s$ is selected as the following:

$$a^\star \leftarrow \arg\max_{a_1, \cdots, a_N} Q(s, a_i), \quad a_1, \cdots, a_N \sim \pi_{f_\beta}(\cdot \mid s). \tag{89}$$

**5. Gaussian.**

**RLPD** (Ball et al., 2023) is a strong offline-to-online RL method that trains a SAC agent from scratch online with a 50/50 sampling scheme (*i.e.*, 50% of training examples in a batch comes from the

---

**Algorithm 1** Learning procedure in QAM.

---

**Input:** $(s, a, s', r)$: off-policy transition tuple, $f_\beta$: behavior velocity field, $f_\theta$ fine-tuned velocity field, $Q_\phi$: critic function.

Optimize $\phi$ w.r.t $L(\phi) = \left[Q_\phi(s, a) - r - \gamma Q_{\bar{\phi}}(s', a' \sim \pi_\theta(\cdot \mid s'))\right]^2$ ▷ *TD backup*

$\boldsymbol{a} = \{a_0, a_h, \cdots, a_1\} \leftarrow \text{SDE}(f_\theta(s, \cdot, \cdot))$ ▷ *Memory-less SDE; Equation* (24)

$\tilde{g}_1 \leftarrow -\tau \nabla_{a_1} Q_\phi(s, a_1)$ ▷ *Compute the critic's action gradient*

$\tilde{g}_0, \tilde{g}_h, \cdots, \tilde{g}_{1-h} \leftarrow \text{LeanAdj}(f_\beta(s, \cdot, \cdot), \tilde{g}_1, \boldsymbol{a})$ ▷ *Lean adjoint states; Equation* (25)

Optimize $\theta$ w.r.t $L(\theta) = \sum_t \left\| \frac{2(f_\theta(s, a_t, t) - f_\beta(s, a_t, t))}{\sigma_t} + \sigma_t \tilde{g}_t \right\|_2^2$ ▷ *Adjoint matching; Equation* (21)

**Output:** $f_\theta, Q_\phi$

---

offline dataset whereas the other $50\%$ of training examples comes from the online replay buffer). In addition to the standard RLPD update, we also add a behavior cloning (BC) loss which we find to improve the performance further. We tune over the BC coefficient ($\alpha$) and report the value we use for each domain in Table 5.

**ReBRAC** (Tarasov et al., 2024) is a strong offline RL method that trains a TD3 (Fujimoto et al., 2018) agent with behavior cloning loss. In practice, we find two hyperparameters to impact performance the most. The first hyperparameter is $\alpha$ which controls the strength of the behavior cloning loss. The second hyperparameter is $\sigma$ which controls the magnitude of the Gaussian noise added in the TD3 policy. We keep the action noise clip to be $0.5$ and an actor delay of $2$, following the original paper.

## F  PSEUDOCODE AND HYPERPARAMETERS

While most methods share a common set of hyperparameters (Table 4 for a fair comparison, most methods need to be tuned for each domain. We include the domain-specific in Table 5 and the tuning range of them in Table 6. To pick the hyperparmaeter for each domain for each method, we first run a sweep over all hyperparmeters in the range (specified by Table 6), or all combinations of them if there are multiple hyperparameters involved. The hyperparameter tuning runs use 4 seeds for each method for each hyperparameter configuration for each of the two tasks per domain. For locomotion domains, we use task 1 (the default task) and task 4. For manipulation domains, we use task 2 (the default task) and task 4. We use task 4 in addition to the default task recommended by OGBench because we often find the combination of these two cover the characteristics of each domain better. We then use the combined performance of the two tuning tasks per domain per method to pick the hyperparameter configuration. We include them in Table 5. Finally, for all our main results, we run all methods on all five tasks for each domain on 12 *new* seeds (different from the tuning seeds). We pick the hyperparmeter range such that the total number of tuning runs are similar across methods. To achieve this, we use the following strategy: For methods where more than one hyperparameter needs to be tuned, we use a coarser hyperparmeter range. For methods where there is only one hyperparmeter, we use a more fine-grained sweep.

## G  THEORETICAL GUARANTEES

---

**Proposition 1** (Extension of Proposition 7 in Domingo-Enrich et al. (2025) to Policy Optimization.) Take $L_{\text{AM}}(\theta)$ in Equation (14), there is a unique $f_\theta$ such that

$$\frac{\partial}{\partial f_\theta} L_{\text{AM}} = 0, \tag{90}$$

and for all $s \in \text{supp}(D)$,

$$\pi_\theta(\cdot \mid s) \propto \pi_\beta(\cdot \mid s) e^{\tau Q_\phi(s, a)}. \tag{91}$$

---

*Proof.* Our proof mainly rewrites the assumptions and the statement of Proposition 7 of Domingo-Enrich et al. (2025) in our notations with a simple extra step that extends it to the state-conditioned version (*e.g.*, for policy optimization).

We first define (from Equation 27 in Domingo-Enrich et al. (2025)) the residual velocity field as

$$f_{\text{res}}(s, a_t, t) := \sqrt{\frac{2}{\beta_t(\beta_t \dot{\alpha}_t/\alpha_t - \dot{\beta}_t)}}(f_\theta(s, a_t, t) - f_\beta(s, a_t, t)). \tag{92}$$

While the original result showed for a more general case with a family of $\alpha, \beta$ (and later on $\sigma$), in our work we assume

$$\alpha(t) = t, \tag{93}$$
$$\beta(t) = 1 - t, \tag{94}$$
$$\sigma(t) = \sqrt{2(1-t)/t}. \tag{95}$$

This allows us to simplify the expression for $f_{\text{res}}$ as

$$f_{\text{res}}(s, a_t, t) = \sqrt{\frac{2t}{1-t}}(f_\theta(s, a_t, t) - f_\beta(s, a_t, t)) \tag{96}$$

Then, we restate the definition of $b$ (from Equation 27 in Domingo-Enrich et al. (2025)):

$$b(s, a_t, t) = 2(f_\beta(s, a_t, t) + f_\theta(s, a_t, t)) - \frac{\dot{\alpha}_t}{\alpha_t}a_t - \sigma(t)f_{\text{res}}(s, a_t, t) \tag{97}$$

$$= 2(f_\beta(s, a_t, t) + f_\theta(s, a_t, t)) - a_t/t - 2f_\theta(s, a_t, t) \tag{98}$$

$$= 2f_\beta(s, a_t, t) - a_t/t \tag{99}$$

We can now rewrite our 'lean' adjoint state definition as

$$\mathrm{d}\tilde{g}_t = -\tilde{g}_t^\top \nabla_{a_t} [b(s, a_t, t)], \quad \tilde{g}_1 = -\tau \nabla_a Q_\phi(s, a_1) \tag{100}$$

which coincides with the definition in Equation 38 in Domingo-Enrich et al. (2025), with $f = 0$ and $g(\cdot) = -\tau Q_\phi(s, \cdot)$. Now, we can rewrite our adjoint matching objective as

$$L_{\text{AM}}(\theta) := \mathbb{E}_{s \sim D, \{a_t\}_t} \left[ \tilde{L}_{\text{AM}}(f_\theta, \{a_t\}_t) \right] \tag{101}$$

where

$$\tilde{L}_{\text{AM}}(s, f_\theta, \{a_t\}_t) := \int_0^1 \| f_{\text{res}}(s, a_t, t) + \sigma(t)\tilde{g}_t \| \, \mathrm{d}t$$

$$= \frac{1}{2} \int_0^1 \left\| \sqrt{\frac{2t}{1-t}} f_\theta(s, a_t, t) + \sigma(t)\tilde{g}_t \right\|_2^2 \mathrm{d}t \tag{102}$$

$$= \frac{1}{2} \int_0^1 \| 2f_\theta(s, a_t, t)/\sigma(t) + \sigma(t)\tilde{g}_t \|_2^2 \, \mathrm{d}t$$

Comparing $\tilde{L}_{\text{AM}}(s, f_\theta, \{a_t\}_t)$ to the definition in Equation 37 in Domingo-Enrich et al. (2025), they are different by only a factor of 2 *conditioned on a fixed $s$*. Thus, their critical points are the same.

By triggering Proposition 7 in Domingo-Enrich et al. (2025), we can conclude that for a fixed $s$, the only critical point of the following loss function,

$$\mathbb{E}_{\{a_t\}_t} \left[ \tilde{L}_{\text{AM}}(s, f_\theta, \{a_t\}_t) \right], \tag{103}$$

is $f^\star(s, a_t, t)$, the velocity field that generates the following distribution,

$$\pi^\star(\cdot \mid s) \propto \pi_\beta(\cdot \mid s) \exp(\tau Q_\phi(s, \cdot)). \tag{104}$$

Finally, since the $L_{\text{AM}}(\theta)$ is a linear combination of $\mathbb{E}_{\{a_t\}_t} \left[ \tilde{L}_{\text{AM}}(s, f_\theta, \{a_t\}_t) \right]$ over different $s$, the critic point of $L_{\text{AM}}(f_\theta)$ is simply the cartesian product of over the critic points for each $s \in \text{supp}(D)$. Since there is only one critical point for each $\mathbb{E}_{\{a_t\}_t} \left[ \tilde{L}_{\text{AM}}(s, f_\theta, \{a_t\}_t) \right]$, $L_{\text{AM}}(f_\theta)$ also has only one critical point and coincides with $f^\star(s, a_t, t)$. This concludes that the only critical point of $L_{\text{AM}}(f_\theta)$ results in velocity fields that satisfy

$$\pi^\star(\cdot \mid s) \propto \pi_\beta(\cdot \mid s) \exp(\tau Q_\phi(s, \cdot)), \quad \forall s \in \text{supp}(D). \tag{105}$$

$\square$

| | | ReBRAC | FBRAC | BAM | FQL | FAWAC | CGQL | CGQL-M | CGQL-L | DAC | QSM | DSRL | FEdit | IFQL | QAM | QAM-F | QAM-E |
|---|---|---|---|---|---|---|---|---|---|---|---|---|---|---|---|---|---|
| antmaze-large | task1 | **98** | 0 | 90 | **93** | 6 | 63 | 56 | 39 | 88 | 87 | 62 | 67 | 41 | 77 | 85 | 87 |
| | task2 | **88** | 0 | 60 | 86 | 1 | 73 | 49 | 51 | 71 | 78 | 75 | 67 | 14 | 80 | 64 | 81 |
| | task3 | **98** | 12 | 94 | 59 | 40 | 91 | 90 | 79 | **98** | 99 | 81 | 65 | 54 | 89 | 96 | 94 |
| | task4 | **94** | 0 | 85 | 54 | 18 | 78 | 78 | 76 | 91 | 91 | 18 | 28 | 25 | 69 | 81 | 70 |
| | task5 | **95** | 0 | 88 | 86 | 22 | 75 | 80 | 78 | 92 | 96 | 69 | 63 | 45 | 89 | 87 | 83 |
| | agg. (5 tasks) | **94** | 2 | 84 | 76 | 17 | 76 | 71 | 65 | 88 | 90 | 61 | 58 | 36 | 81 | 83 | 83 |
| antmaze-giant | task1 | 36 | 0 | 3 | 0 | 0 | 2 | 0 | 0 | 53 | **62** | 0 | 0 | 0 | 7 | 11 | 0 |
| | task2 | **73** | 0 | 0 | 0 | 0 | 0 | 0 | 4 | 0 | 1 | 0 | 0 | 0 | 0 | 0 | 0 |
| | task3 | **13** | 0 | 0 | 0 | 0 | 0 | 0 | 0 | 0 | 0 | 0 | 0 | 1 | 0 | 0 | 0 |
| | task4 | **74** | 0 | 0 | 0 | 0 | 0 | 0 | 0 | 0 | 0 | 12 | 10 | 2 | 34 | 5 | 0 |
| | task5 | **89** | 0 | 1 | 0 | 0 | 0 | 22 | 11 | 25 | 58 | 2 | 0 | 2 | 49 | 43 | 6 |
| | agg. (5 tasks) | **57** | 0 | 1 | 0 | 0 | 0 | 4 | 3 | 16 | 24 | 3 | 2 | 1 | 18 | 12 | 1 |
| humanoidmaze-medium | task1 | 38 | 26 | 49 | 34 | 18 | 30 | 8 | 55 | 87 | **88** | 49 | 0 | 86 | 40 | 32 | 27 |
| | task2 | 91 | 78 | 69 | 95 | 44 | 78 | **99** | 93 | 96 | 96 | 91 | 39 | 92 | 97 | 98 | 99 |
| | task3 | 83 | 28 | 75 | **96** | 20 | 78 | 0 | 62 | 92 | 95 | 36 | 0 | 93 | 96 | 95 | 68 |
| | task4 | 37 | 3 | 22 | 14 | 1 | 23 | 0 | 2 | 45 | 31 | 0 | 0 | **60** | 3 | 0 | 0 |
| | task5 | 96 | 50 | 83 | **99** | 36 | 89 | 100 | 98 | 99 | 98 | 90 | 68 | 98 | 99 | 99 | 99 |
| | agg. (5 tasks) | 69 | 39 | 60 | 68 | 24 | 60 | 42 | 62 | 83 | 82 | 53 | 22 | **86** | 67 | 65 | 59 |
| humanoidmaze-large | task1 | **36** | 0 | 5 | 8 | 0 | 2 | 0 | 2 | 0 | 10 | 14 | 8 | **36** | 6 | 8 | 6 |
| | task2 | **1** | 0 | 0 | 0 | 0 | 0 | 0 | 0 | 0 | 0 | 0 | 0 | 0 | 0 | 0 | 0 |
| | task3 | 32 | 0 | 6 | 19 | 1 | 11 | 4 | 18 | 0 | 16 | 1 | 3 | **55** | 16 | 19 | 4 |
| | task4 | **10** | 0 | 6 | 9 | 0 | 5 | 8 | 4 | 0 | 2 | 0 | 1 | 2 | 16 | 12 | 0 |
| | task5 | 7 | 0 | **11** | 9 | 0 | 5 | 16 | 9 | 1 | 2 | 2 | 1 | 29 | 19 | 19 | 0 |
| | agg. (5 tasks) | 17 | 0 | 5 | 9 | 0 | 5 | 6 | 6 | 0 | 6 | 3 | 3 | **24** | 11 | 12 | 2 |
| scene-sparse | task1 | 98 | 66 | **100** | 99 | 62 | 79 | **100** | **100** | **100** | **100** | **100** | 95 | 93 | **100** | **100** | **100** |
| | task2 | 90 | 80 | 99 | 71 | 15 | 88 | 99 | 98 | 99 | 78 | **100** | 97 | 63 | 98 | 99 | 98 |
| | task3 | 51 | 41 | 99 | 97 | 14 | 23 | 92 | 91 | 79 | 97 | 97 | 61 | 71 | **100** | 95 | **100** |
| | task4 | 60 | 49 | 96 | 92 | 71 | 0 | 6 | 86 | 8 | 92 | **100** | 35 | 98 | **100** | 93 | **100** |
| | task5 | 27 | 17 | 96 | 33 | 27 | 0 | 74 | 66 | 53 | 25 | **100** | 24 | 95 | 87 | 88 | 88 |
| | agg. (5 tasks) | 65 | 50 | 98 | 78 | 38 | 38 | 74 | 88 | 68 | 78 | **99** | 62 | 84 | 97 | 95 | 97 |
| puzzle-3x3-sparse | task1 | 99 | 1 | 26 | 99 | 8 | 87 | **100** | 98 | 98 | 89 | 83 | **100** | **100** | 98 | 98 | **100** |
| | task2 | 77 | 0 | 75 | 79 | 2 | 55 | **100** | 92 | 66 | 85 | 83 | **100** | **100** | **100** | **100** | **100** |
| | task3 | 85 | 0 | 78 | 83 | 1 | 24 | **100** | 87 | 54 | 18 | 83 | 98 | **100** | **100** | **100** | **100** |
| | task4 | 62 | 0 | 92 | 84 | 1 | 25 | **100** | 85 | 72 | 87 | 83 | 97 | **100** | **100** | **100** | **100** |
| | task5 | 70 | 1 | 9 | 6 | 1 | 47 | **100** | 88 | 51 | 9 | **100** | **100** | **100** | **100** | **100** | **100** |
| | agg. (5 tasks) | 79 | 0 | 56 | 70 | 3 | 48 | **100** | 90 | 68 | 57 | 87 | 99 | **100** | **100** | 99 | **100** |
| puzzle-4x4-100M-sparse | task1 | 0 | 29 | 0 | 17 | 0 | 56 | 0 | 0 | 0 | 0 | 0 | 71 | 0 | 0 | 20 | **85** |
| | task2 | 0 | 13 | 0 | 1 | 0 | 12 | 0 | 0 | 0 | 0 | 1 | 13 | 1 | 0 | 5 | 7 |
| | task3 | 0 | 16 | 0 | 3 | 0 | 41 | 0 | 0 | 0 | 0 | 0 | 39 | 0 | 0 | 3 | **56** |
| | task4 | 0 | 6 | 0 | 2 | 0 | 11 | 0 | 0 | 0 | 0 | 0 | 14 | 0 | 0 | 4 | **29** |
| | task5 | 0 | 13 | 0 | 2 | 0 | 3 | 0 | 0 | 0 | 0 | 0 | 32 | 0 | 0 | 2 | 19 |
| | agg. (5 tasks) | 0 | 15 | 0 | 5 | 0 | 24 | 0 | 0 | 0 | 0 | 0 | 34 | 0 | 0 | 6 | **39** |
| cube-double | task1 | 29 | 0 | 84 | 81 | 8 | 55 | 50 | 62 | 36 | 80 | **90** | 77 | 16 | 85 | 84 | 89 |
| | task2 | 6 | 0 | 49 | 46 | 0 | 39 | 46 | 52 | 35 | 29 | **87** | 28 | 12 | 79 | 84 | 77 |
| | task3 | 2 | 0 | 38 | 42 | 0 | 44 | 50 | 52 | 31 | 33 | **85** | 44 | 10 | 54 | 59 | 55 |
| | task4 | 1 | 0 | 8 | 10 | 0 | 13 | 11 | 18 | 16 | 5 | **32** | 14 | 4 | 22 | 18 | 22 |
| | task5 | 4 | 0 | 56 | 50 | 0 | 42 | 48 | 41 | 56 | 16 | 76 | 39 | 11 | 82 | 81 | **83** |
| | agg. (5 tasks) | 9 | 0 | 47 | 46 | 2 | 38 | 41 | 45 | 35 | 33 | **74** | 40 | 11 | 64 | 65 | 65 |
| cube-triple | task1 | 4 | 2 | 14 | 15 | 0 | 39 | 40 | **40** | 24 | 26 | 7 | 11 | 2 | 14 | 11 | 16 |
| | task2 | 0 | 0 | 0 | 0 | 0 | 0 | 0 | 0 | 0 | 1 | 0 | 0 | 0 | 1 | 0 | 2 |
| | task3 | 0 | 0 | 3 | 1 | 0 | 1 | 0 | 0 | 0 | 0 | 0 | 0 | 0 | 2 | 2 | 2 |
| | task4 | 0 | 0 | 0 | 0 | 0 | 0 | 0 | 0 | 0 | 0 | 0 | 0 | 0 | 0 | 0 | 2 |
| | task5 | 0 | 0 | 0 | 0 | 0 | 0 | 0 | 0 | 0 | 0 | 0 | 0 | 0 | 0 | 0 | 0 |
| | agg. (5 tasks) | 1 | 0 | 3 | 3 | 0 | 8 | 8 | 8 | 5 | 6 | 1 | 2 | 0 | 3 | 3 | 5 |
| cube-quadruple-100M | task1 | 35 | 0 | 0 | 11 | 0 | 0 | 2 | 2 | 12 | **90** | 9 | 19 | 8 | 13 | 67 | 32 |
| | task2 | 6 | 0 | 0 | 0 | 0 | 0 | 0 | 0 | 1 | 0 | 1 | 2 | 0 | 0 | 1 | 0 |
| | task3 | 3 | 0 | 0 | 0 | 0 | 0 | 0 | 0 | 0 | 0 | 1 | 2 | 2 | 1 | 3 | 0 |
| | task4 | 0 | 0 | 0 | 0 | 0 | 0 | 0 | 0 | 0 | 4 | 0 | 1 | 0 | 0 | 0 | 0 |
| | task5 | 0 | 0 | 0 | 0 | 0 | 0 | 0 | 0 | 0 | 0 | 0 | 0 | 0 | 0 | 0 | 0 |
| | agg. (5 tasks) | 9 | 0 | 0 | 2 | 0 | 0 | 1 | 0 | 3 | **19** | 2 | 5 | 2 | 3 | 14 | 6 |
| all | agg. (50 tasks) | 40 | 11 | 35 | 36 | 8 | 30 | 35 | 37 | 36 | 39 | 38 | 33 | 34 | 44 | 45 | **46** |

Table 2: **Full offline results at 1M training steps (12 seeds).**

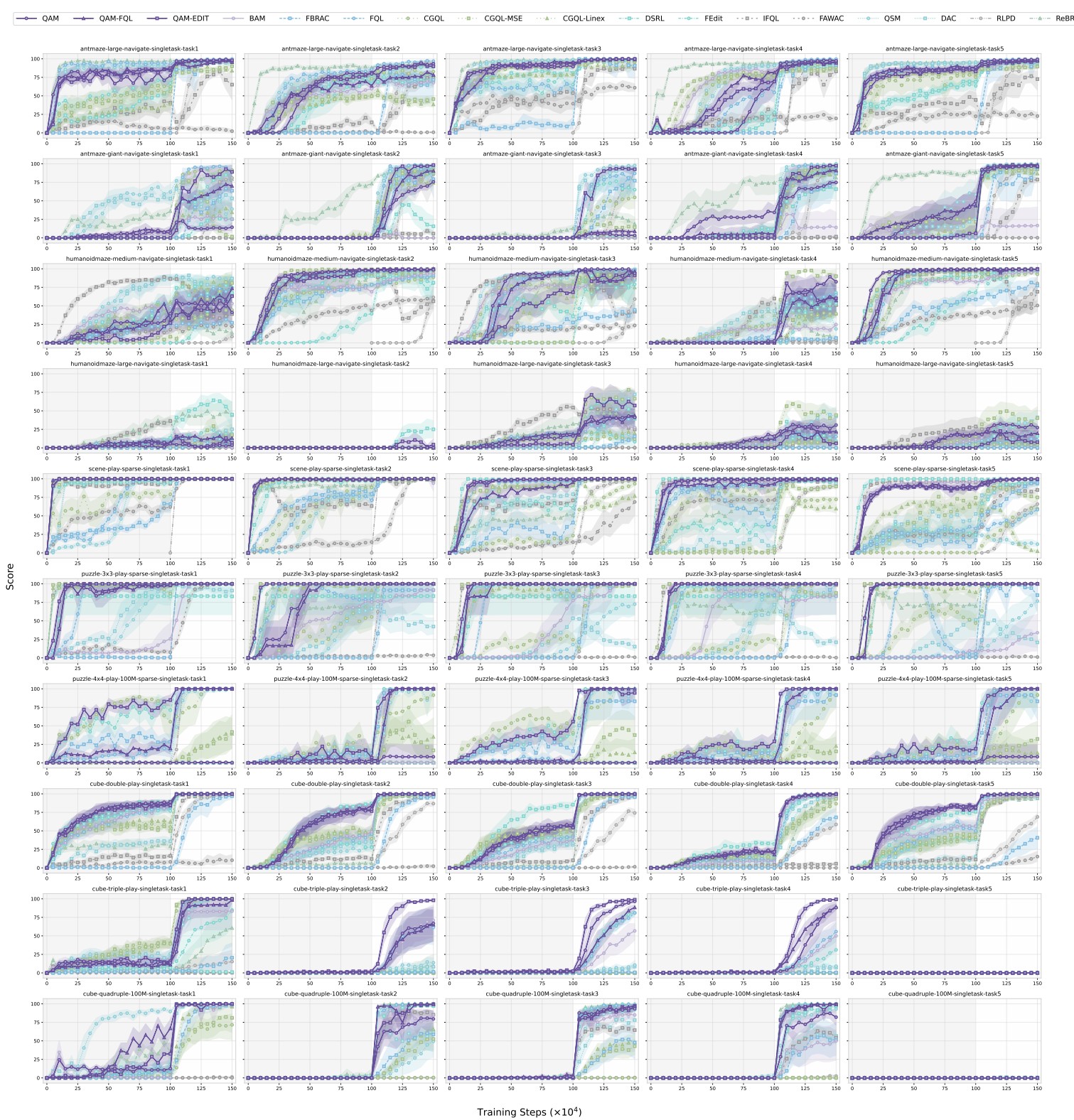

Figure 5: **Full training curves for our main results (12 seeds).**

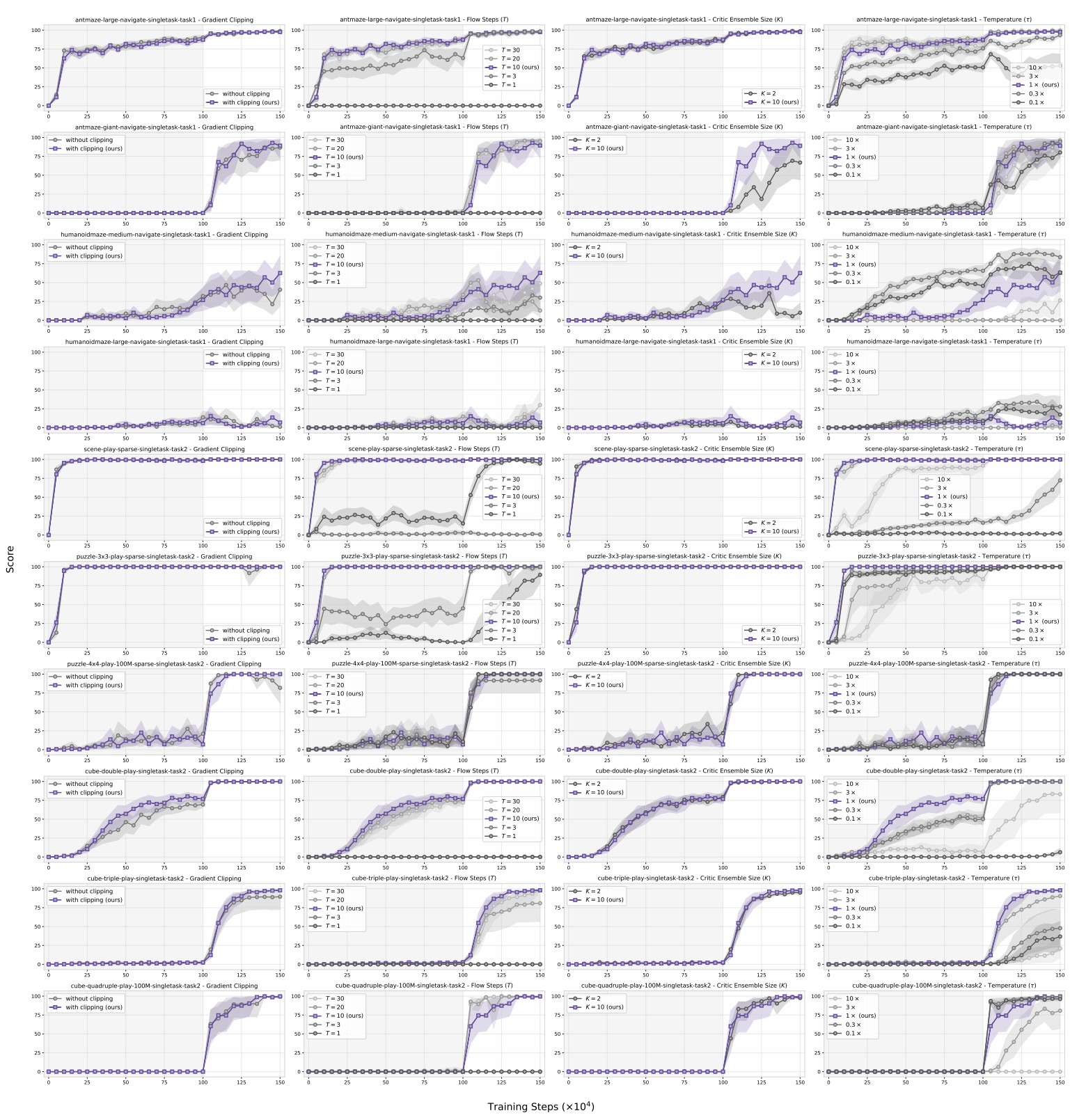

Figure 6: **Full sensitivity results for** `QAM-E` **(12 seeds).**

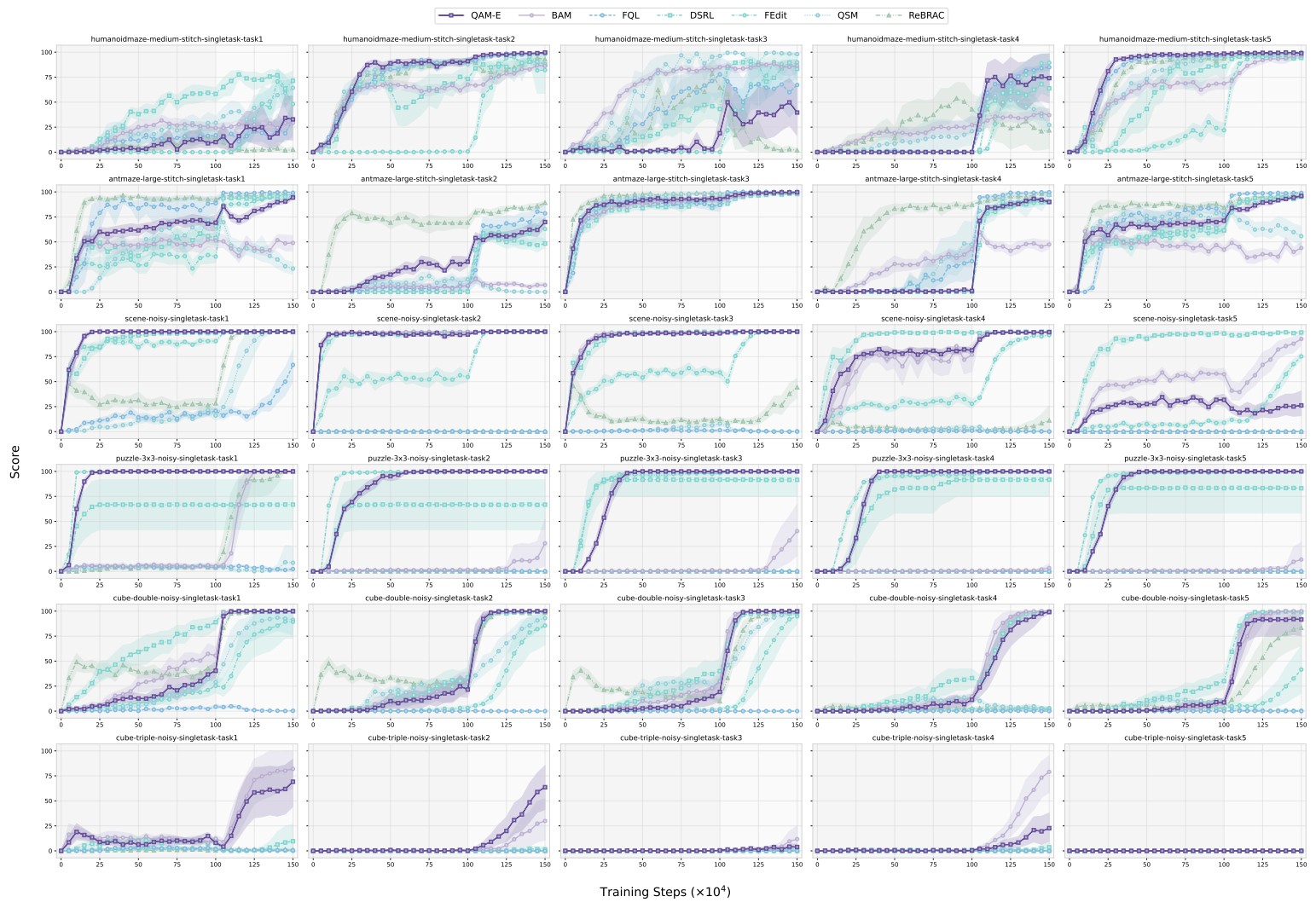

Figure 7: **Full training curves for data quality analysis (12 seeds).**

| Tasks | Dataset Size | Episode Length | Action Dimension ($A$) |
|---|---|---|---|
| `cube-double-*` | 1M | 500 | 5 |
| `cube-triple-*` | 3M | 1000 | 5 |
| `cube-quadruple-100M-*` | 100M | 1000 | 5 |
| `antmaze-large-*` | 1M | 1000 | 8 |
| `antmaze-giant-*` | 1M | 1000 | 8 |
| `humanoidmaze-medium-*` | 2M | 2000 | 21 |
| `humanoidmaze-large-*` | 2M | 2000 | 21 |
| `scene-sparse-*` | 1M | 750 | 5 |
| `puzzle-3x3-sparse-*` | 1M | 500 | 5 |
| `puzzle-4x4-100M-sparse-*` | 100M | 500 | 5 |

Table 3: **Domain metadata.**

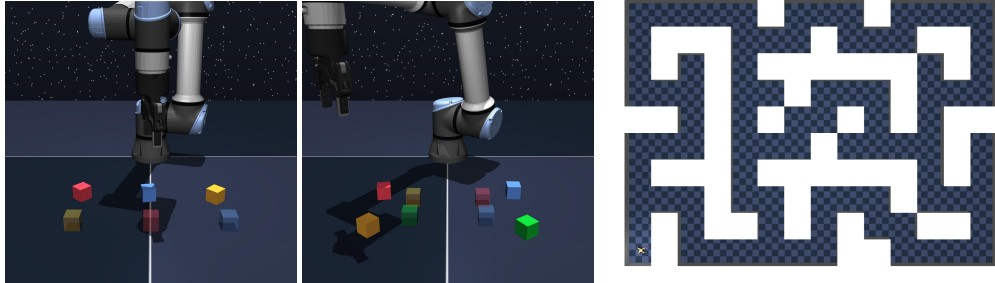

Figure 8: **OGBench domains**. In this paper, we primarily focus on evaluating our method on some of the hardest domains on OGBench (Park et al., 2025a). cube-{triple, quadruple} (left) requires a robot arm to manipulate up to 3/4 cubes from an initial arrangement to a goal arrangement. antmaze-giant (right) requires an ant robot agent to navigate from one location to another location. All of these domains are long-horizon by design and are difficult to solve from offline data alone. The offline dataset in these domains contain diverse multi-modal behaviors that can only be accurately captured by expressive generative models like flow/diffusion policies.

| Parameter | Value |
|---|---|
| Batch size | 256 |
| Discount factor ($\gamma$) | 0.99 for {puzzle/scene/cube/antmaze-large}-* 
 0.995 for {humanoidmaze/antmaze-giant}-* |
| Optimizer | Adam |
| Learning rate | $3 \times 10^{-4}$ |
| Target network update rate ($\lambda$) | $5 \times 10^{-3}$ |
| Critic ensemble size ($K$) | 10 |
| Critic target pessimistic coefficient ($\rho$) | 0.5 for {puzzle/scene/cube/antmaze}-* 
 0 for humanoidmaze-* |
| UTD ratio | 1 |
| Number of flow steps ($T$) | 10 |
| Number of offline training steps | $10^6$; except RLPD (0) |
| Number of online environment steps | $0.5 \times 10^6$ |
| Network width | 512 |
| Network depth | 4 hidden layers |
| Optimizer gradient max norm clipping | 1 for QSM, DAC, BAM and QAM*. No clipping for others. |

Table 4: **Common hyperparameters.**

| Domains | RLPD $\alpha$ | ReBRAC $(\alpha,\sigma)$ | FAWAC $\tau$ | QSM $(\tau,\eta)$ | DAC $\eta$ | FBRAC $\alpha$ | FQL $\alpha$ | DSRL $\sigma_z$ | FEdit $\sigma_a$ | IFQL $\tau$ | CGQL $(\vartheta,\tau)$ | CGQL-M $(\vartheta,\varrho,\tau)$ | CGQL-L $(\vartheta,\varrho,\tau)$ | BAM $\tau$ | QAM $\tau$ | QAM-E $(\tau,\sigma_a)$ | QAM-F $(\tau,\alpha)$ |
|---|---|---|---|---|---|---|---|---|---|---|---|---|---|---|---|---|---|
| scene-sparse-* | 0.3 | (0.03,0) | 6.4 | (3,30) | 1 | 100 | 300 | 0.4 | 0.2 | 0.9 | (10,0.1) | (10,0.1,0.1) | (10,0.1,0.1) | 3 | 1 | (1,0) | (1,300) |
| puzzle-3x3-sparse-* | 0.01 | (0.1,0) | 0.8 | (3,30) | 1 | 0.3 | 300 | 1 | 0.2 | 0.95 | (10,0.1) | (10,0.1,0.1) | (10,0.001,0.1) | 10 | 3 | (1,0.1) | (3,∞) |
| puzzle-4x4-100M-sparse-* | 0 | (0.01,0.2) | 0.8 | (10,1) | 1 | 0.3 | 1 | 1 | 0.8 | 0.9 | (10,0.1) | (10,0.001,1) | (10,0.001,1) | 30 | 30 | (0.1,0.9) | (3,3) |
| cube-double-* | 0.1 | (0.01,0) | 0.8 | (3,10) | 3 | 0.1 | 300 | 1 | 0.2 | 0.9 | (10,0.01) | (10,0.001,0.01) | (10,0.001,0.01) | 3 | 1 | (1,0) | (1,∞) |
| cube-triple-* | 0 | (0.01,0.2) | 0.8 | (3,3) | 0.3 | 0.03 | 30 | 1.4 | 0.3 | 0.95 | (10,0.1) | (10,0.001,0.1) | (10,0.001,0.1) | 30 | 3 | (3,0.1) | (10,300) |
| cube-quadruple-100M-* | 0 | (0.01,0.2) | 0.8 | (1,10) | 1 | 1 | 100 | 1.4 | 0.4 | 0.95 | (10,0.1) | (10,0.1,0.1) | (10,0.1,0.1) | 10 | 1 | (3,0.1) | (0.3,30) |
| antmaze-large-* | 0 | (0.01,0) | 6.4 | (30,30) | 0.3 | 0.1 | 3 | 0.8 | 0.2 | 0.9 | (10,0.1) | (10,0.1,0.1) | (10,0.001,0.1) | 30 | 10 | (1,0.1) | (3,30) |
| antmaze-giant-* | 0.01 | (0.01,0) | 0.8 | (30,30) | 0.3 | 0.1 | 3 | 1.2 | 0.3 | 0.8 | (10,0.1) | (10,0.001,0.1) | (10,0.001,0.1) | 10 | 3 | (10,0.1) | (3,30) |
| humanoidmaze-medium-* | 0.03 | (0.01,0) | 6.4 | (10,30) | 1 | 30 | 30 | 0.6 | 0.5 | 0.7 | (10,0.01) | (10,0.1,0.1) | (10,0.1,0.1) | 10 | 3 | (3,0.1) | (1,30) |
| humanoidmaze-large-* | 0 | (0.01,0.1) | 0.8 | (10,30) | 0.3 | 10 | 30 | 0.8 | 0.1 | 0.8 | (10,0.01) | (10,0.1,0.1) | (10,0.1,0.1) | 10 | 3 | (3,0.1) | (0.3,30) |

Table 5: **Domain-specific hyperparameters.** The best hyperparameter configuration obtained from our tuning runs. We use the same hyperparameter configuration for all tasks in each domain.

| Method | Hyperparameter(s) | Sweep Range |
|---|---|---|
| RLPD | $\alpha$ | $\{0, 0.003, 0.01, 0.03, 0.1, 0.3, 1, 3, 10, 30\}$ |
| ReBRAC | $(\alpha, \sigma)$ | $(\{0.01, 0.03, 0.1, 0.3, 1\}, \{0, 0.1, 0.2\})$ |
| FBRAC | $\alpha$ | $\{0.03, 0.1, 0.3, 1.0, 3.0, 10.0, 30.0, 100.0\}$ |
| CGQL | $(\vartheta, \tau)$ | $(\{0.01, 0.1, 1\}, \{0.1, 1, 10\})$ |
| CGQL-{M,L} | $(\vartheta, \tau, \varrho)$ | $(\{0.01, 0.1, 1\}, \{0.1, 1, 10\}, \{0.001, 0.1\})$ |
| FQL | $\alpha$ | $\{0.3, 1, 3, 10, 30, 100, 300, 1000\}$ |
| DSRL | $\sigma_z$ | $\{0.1, 0.2, 0.4, 0.6, 0.8, 1, 1.2, 1.4\}$ |
| FEdit | $\sigma_a$ | $\{0.1, 0.2, 0.3, 0.4, 0.5, 0.6, 0.7, 0.8\}$ |
| FAWAC | $\tau$ | $\{0.1, 0.2, 0.4, 0.8, 1.6, 3.2, 6.4, 12.8\}$ |
| IFQL | $\kappa$ | $\{0.5, 0.6, 0.7, 0.8, 0.9, 0.95\}$ |
| DAC | $\eta$ | $\{0.1, 0.3, 1, 3.0, 10, 30, 100, 300\}$ |
| QSM | $(\tau, \eta)$ | $(\{1.0, 3.0, 10.0, 30\}, \{1, 3, 10, 30\})$ |
| BAM | $\tau$ | $\{0.1, 0.3, 1, 3, 10, 30\}$ |
| QAM | $\tau$ | $\{0.1, 0.3, 1, 3, 10, 30\}$ |
| QAM-E | $(\tau, \sigma_a)$ | $(\{0.1, 0.3, 1, 3, 10\}, \{0, 0.1, 0.5, 0.9\})$ |
| QAM-F | $(\tau, \alpha)$ | $(\{0.1, 0.3, 1, 3, 10\}, \{3, 30, 300, \infty\})$ |

Table 6: **Domain-specific hyperparameter tuning range.** For QAM-F, $\alpha = \infty$ is equivalent to the original QAM. Similarly, for QAM-E, $\sigma_a = 0$ is equivalent to the original QAM. For methods with more than one hyperparameter, we tune all possible combinations within the specified sweep ranges. For example, for CGQL, we sweep over all $3 \times 3 = 9$ configurations with 3 possible values of $\vartheta$ and 3 possible values of $\tau$.

| Method | Training Speed (milliseconds/step) | Parameter Count |
|---|---|---|
| ReBRAC | 2.97 | 9 226 271 |
| FBRAC | 4.54 | 9 237 535 |
| CGQL | 9.62 | 9 242 655 |
| CGQL-{M,L} | 10.40 | 9 242 655 |
| FQL | 3.58 | 10 082 868 |
| DSRL | 5.63 | 18 474 580 |
| FEdit | 3.77 | 10 093 642 |
| FAWAC | 3.44 | 10 065 952 |
| IFQL | 2.66 | 10 065 952 |
| DAC, QSM | 4.44 | 10 320 447 |
| BAM | 5.52 | 10 083 380 |
| QAM | 5.83 | 10 083 380 |
| QAM-E | 6.61 | 10 939 487 |
| QAM-F | 6.36 | 10 928 713 |

Table 7: **Training speed and parameter count for each method on** cube-triple.

