# OpenReview forum: "Q-Learning with Adjoint Matching"
_ICLR.cc/2026/Conference — ICLR 2026 Poster_

### Official Review · Reviewer_YuTk · 2025-10-28

**Soundness:** 2
**Presentation:** 3
**Contribution:** 3
**Rating:** 4
**Confidence:** 4

**Summary:**

This paper proposes Q-learning with Adjoint Matching (QAM), a novel temporal-difference (TD) based reinforcement learning algorithm that addresses the challenge of optimizing expressive flow/diffusion policies with respect to a parameterized critic function. The key innovation is leveraging adjoint matching, a technique from generative modeling, to utilize the critic's action gradient directly, thereby avoiding the numerical instability that arises from backpropagation through multi-step denoising processes.

**Strengths:**

1. Novel and well-motivated approach: The paper addresses a fundamental tension in continuous-action RL between policy expressivity and optimization tractability.
2. Strong theoretical foundation: The method builds on solid mathematical principles from stochastic optimal control and adjoint methods.
3. Comprehensive experimental evaluation: The paper includes extensive comparisons against eight representative baselines across multiple challenging domains. The categorization of baselines (DDPG-based, value-only, gradient-with-approximations, post-processing, Gaussian) is thoughtful and covers the landscape of existing approaches well.
4. Practical considerations: The authors provide detailed implementation guidance, including discrete approximations, gradient clipping, and ensemble critic training.
5. Clear presentation: The paper is generally well-written with good motivation, clear problem formulation, and helpful visual aids.

**Weaknesses:**

1. Insufficient stability analysis in experiments: Although the success rate is reported, the paper lacks metrics that directly assess stability, such as standard deviation across seeds or training loss curves. These would provide concrete evidence of the claimed stability advantages.
2. Limited domain diversity: All evaluated tasks are from OGBench with similar characteristics (long-horizon, sparse rewards). The generalizability to dense reward tasks (e.g., MuJoCo locomotion in D4RL), tasks with different action space dimensions, shorter horizon tasks, and continuous reward settings remains unclear. This significantly limits the assessment of the method's broad applicability.
3. Impact on Q-value estimation: The paper does not analyze how QAM affects the critic learning process. Since the policy used for target value backup changes during training, this could introduce additional variance or bias in TD targets. An empirical analysis that shows Q-value estimation error over training, compares learned Q-values with Monte Carlo returns, and examines the impact on critic ensemble disagreement would provide important insights.
4. Incomplete sensitivity analysis: The impact of ensemble size and pessimistic coefficient is not studied, despite being set to specific values. No ablation on the choice of "memoryless" SDE vs. alternative noise schedules. The interaction between temperature and clipping is not explored.
5. CGQL is presented as a "novel baseline" without prior work validation. Several baselines (DSRL, FEdit) have been modified from their original implementations, which may not represent their optimal performance. The choice to disable best-of-N sampling for IFQL may unfairly handicap it.
6. Missing ablations: What happens if using the "basic" adjoint matching (Eq. 12) instead of the "lean" version (Eq. 14)? How does QAM perform with different critic architectures or learning rates?
7. Limited theoretical analysis of stability: While the paper claims that adjoint matching avoids instability by not backpropagating through the residual flow model, the theoretical justification is largely intuitive. The discussion on pages 5-6 about ill-conditioned gradients is informal. A rigorous stability analysis (e.g., condition number bounds, gradient variance analysis, or convergence rate guarantees) would significantly strengthen the claims.

**Questions:**

Please see Weaknesses.

---

> ### Author Response · Authors · 2025-11-21
>
> Thanks for your review and your review has already resulted in various improvements to our paper! We are glad to hear that you found our paper well-written, our method well-motivated and novel, and our experiments comprehensive and thoughtful across multiple challenging domains. We are also glad that you appreciate the details we include on implementation detail guidance, gradient clipping and ensemble critic training.
>
> For this rebuttal, **(1) we address your concern about stability analysis by pointing to empirical evidence demonstrating the stability of our method; (2) we address the lack of theoretical analysis by adding a concrete theoretical result establishing that the critical point of our objective coincides with the optimal behavior regularized policy; (3) we address your concerns on the baseline validity by clarifying a few misunderstandings; (4) we also add a missing ablation, basic adjoint matching (BAM) and showed that QAM outperformed BAM consistently across the board (in Table 1); (5) we expand our sensitivity analysis on more hyperparameters and diversify our experiments by demonstrating QAM’s robustness to data corruptions**
>
> ---
>
> ## 1. Insufficient stability analysis
>
> Our original submission already included direct evidence of stability (e.g., training curves in Figure 2). With new experiments in the rebuttal (**on 9x more tasks**), we would like to point the reviewer to our updated Figure 2 which shows online fine-tuning curves across different domains. **Our method, QAM, exhibits a low variance across seeds (as indicated by the very narrow confidence interval) across all domains.**
>
> In contrast, baselines such as FBRAC, which use backpropagation through time, exhibit much larger variance across seeds (e.g., on puzzle-4x4, antmaze-large, and cube-quadruple). **QAM is also the most robust across the board and notably it outperforms all other methods on the cube-triple domain (aggregated over 5 tasks).**
>
> ---
>
> ## 2. Limited theoretical analysis
>
> We added a new theoretical justification (Proposition 1 on Page 7) showing that **the critical point of the adjoint-matching loss coincides with the optimal behavior-regularized policy**:
>
> $$ \pi^\star \propto \pi_\beta(\cdot \mid s) e^{\tau Q(s, a)}. $$
>
> This is a significant result because it implies that **if the adjoint-matching objective is optimized to convergence (ignoring optimization noise and model mismatch), the resulting flow-matching policy exactly recovers the optimal behavior-regularized policy**.
>
> ---
>
> ## 3. Limited domain diversity
>
> To expand the diversity of our empirical evaluations, we conducted additional experiments on datasets with action noises.
>
> In particular, we take two of the hardest tasks in the benchmark together with the noisy-style datasets from OGBench. The noisy-style datasets were collected by expert policies with larger, uncorrelated Gaussian noise, resulting in broader state coverage but larger learning challenge.
>
> We evaluated QAM alongside the strongest baselines (FEdit, FQL, ReBRAC). Results are shown in Figure 6 with the performance at 250K environment steps summarized in the table below (6 seeds). **Our method exhibits strong robustness to action noises: while baselines collapse almost completely on the noisy datasets, QAM only experiences a modest performance drop.**
>
> | Method  | cube-double-task4 (play) | cube-double-task4 (noisy) | cube-triple-task4 (play) | cube-triple-task4 (noisy) |
> |-|-|-|-|-|
> | ReBRAC | 85[77,93] | 4[0,10] | 1[0,1] | 0[0,0] |
> | FEdit | 98[97,99] | 2[1,3] | 1[1,2] | 0[0,1] |
> | FQL | 96[92,99] | 0[0,0] | 32[26,37] | 0[0,0] |
> | QAM | 95[90,98] | 76[40,99] | 70[59,80] | 52[42,64] |
>
> ---
>
> **EDIT: 11/24/2025 -- we managed to run the same analysis on 14x more tasks (see Figure 7 in our updated PDF). While the overall trend stays the same, we would like to highlight that all baselines fail completely on cube-triple environments (across all 5 tasks) and QAM only experiences some performance drop.**

---

> ### Author Response · Authors · 2025-11-22
>
> ## 4. Concerns on the validity of the baselines
>
> Our paper focuses on finding the best policy extraction method for flow/diffusion models. We made our best attempts to make modifications to improve upon their original implementations as they were not designed for OGBench. The modifications are necessary as otherwise these methods are at a disadvantage of not being tuned on the OGBench tasks that we focus on, which are often more difficult to solve.
>
> For example, **our implementation of DSRL matches the reported performance in the original paper on most domains and even outperforms the original implementation on cube-double (e.g., 59 vs. 53), scene (e.g., 100 vs. 88), and puzzle-3x3 (e.g., 80 vs. 0)**. Note that for scene, puzzle-3x3 and puzzle-4x4, we also use the sparse reward formulation, making them more difficult than the default setting (where the agent receives reward based on how much progress it makes in the environment).
>
> For IFQL, we would like to clarify that **we do not disable best-of-N sampling** because it is only done at the evaluation time. We only disable best-of-N sampling for TD backup (which is not done in IFQL as it uses expectile backup) because it would make our experiments too expensive (as we are effectively increasing the batch size for policy sampling by a factor of $N$) and best-of-N scaling is orthogonal to the focus of our work.
>
> For CGQL, we include this baseline because classifier guidance is one of the most natural ideas that one would do when optimizing diffusion/flow models towards some objective functions. We feel obligated to include such a baseline for a comprehensive comparison. For this rebuttal, we additionally include two variants of CGQL for more thorough empirical evaluations: CGQL-MSE/Linex. Conceptually, CGQL uses $\nabla\_a Q(s, a\_1)$ to approximate $\log \mathbb{E}\_{p(a_u \mid a\_1)}\left[\exp(Q(s, a\_1))\right]$ which maybe inaccurate. CGQL-MSE approximates it by training a time-conditioned critic to predict $\mathbb{E}\_{p(a\_u \mid a\_1)}\left[Q(s, a\_1)\right]$, directly dropping the $\log$ and $\exp$. Omitting $\log$ and $\exp$ may also incur additional approximation errors. Thus, we developped CGQL-Linex, which uses the Linex loss to approximate $\log \mathbb{E}\_{p(a\_u \mid a\_1)}\left[\exp(Q(s, a\_1))\right]$ directly without dropping the $\log$ and $\exp$. We refer more details and discussions on these baselines in Appendix C.1. Empirically, we do find that the most principled objective (CGQL-Linex) performs the best with an aggregated score of 34 (vs. CGQL: 27), matching the best flow method, FQL, which obtains an aggregated score of 34.
>
> Overall, we made our best efforts to make sure all methods are evaluated in the most fair way. We follow a **rigorous hyperparameter tuning protocol**:
> - For each method, we tuned hyperparameters using four seeds (1001, 2002, 3003, 4004) on two of the 5 tasks per domain, using **comparable hyperparameter search set cardinalities** to mitigate over-tuning bias (see Table 5 for the hyperparameter sweep ranges).
> - We then evaluated the **best hyperparameter configuration using 8 new seeds** (10001, 20002, 30003, 40004, 50005, 60006, 70007, 80008) across all five tasks in each domain.
>
> We also attach the updated offline RL results below for convenience.
>
> |Category | Method | Aggregated Score (50 tasks) |
> |-|-|-|
> | Gaussian | ReBRAC | 34 |
> | Backprop. | FBRAC | 9 |
> | Backprop. | BAM | 11 |
> | Backprop. | FQL | 34 |
> | Adv. Weighted | FAWAC | 8 |
> | Guidance | CGQL | 27 |
> | Guidance | CGQL-MSE | 33 |
> | Guidance | CGQL-Linex | 34 |
> | Guidance | DAC | 26 |
> | Guidance | QSM | 32 |
> | Post-processing | DSRL | 32 |
> | Post-processing | FEdit | 28 |
> | Post-processing | IFQL | 29 |
> | Adjoint Matching (ours) | QAM | 38 |
> | Adjoint Matching (ours) | QAM-FQL | **41** |
> | Adjoint Matching (ours) | QAM-EDIT | **42** |
>
> ---
>
> EDIT (11/24/2025): The experiments have been completed for all seeds. We have updated the table above to reflect the results with 8 seeds.

---

> ### Author Response · Authors · 2025-11-22
>
> ## 5. Lack of stability analysis and missing ablations on basic adjoint matching.
>
> This is a great point! In this rebuttal, we add basic adjoint matching (BAM) baseline in our experiments to study the stability benefits of our adjoint-matching objective. BAM admits the same optimal solution as QAM but does not use “lean adjoint” states, and thus contains extra terms that incurs additional variance during training (see a more comprehensive discussion in Section 5.2 of [1]).
>
> As we expected, **QAM significantly outperforms BAM across all tasks, providing evidence that lean-adjoint is crucial for stability and performance** (see Table 1 or the markdown table above).
>
> [1] Domingo-Enrich, Carles, et al. "Adjoint matching: Fine-tuning flow and diffusion generative models with memoryless stochastic optimal control." arXiv preprint arXiv:2409.08861 (2024).
>
> ---
>
> ## 6. Sensitivity analysis
>
> We expanded our sensitivity analysis, studying the influence of following components in our algorithm (Figure 3):
> - pessimistic backup coefficient ($\rho$),
> - gradient clipping,
> - number of flow steps ($T$),
> - critic ensemble size ($K$), and finally
> - temperature coefficient ($\tau$)
>
> From our analysis, we found that all of these components contribute to QAM's effectiveness. Among them, the pessimistic backup coefficient, and the temperature parameter ($\tau$) have the biggest impact on QAM’s performance and need to be tuned. In our experiments we use $\tau=0$ for humanoidmaze domains and $\tau=0.5$ for other domains.
>
> Thanks again for your review. Your review has already resulted in various improvements to our paper. **If we have successfully addressed all your concerns, could you kindly update the score?** Please also let us know if you have any additional questions or concerns regarding our paper!

---

> > ### Comment · Reviewer_YuTk · 2025-11-26
> >
> > Thank the authors for their diligent revisions, which have strengthened the paper and addressed most of my concerns. Therefore, I will raise the score to 6.

---

> > > ### Author Response · Authors · 2025-11-26
> > >
> > > Thank you for raising the score and again for your review. Please let us know if there are any remaining concerns or questions that we can address!!

---

### Official Review · Reviewer_V3Vy · 2025-10-29

**Soundness:** 2
**Presentation:** 3
**Contribution:** 3
**Rating:** 4
**Confidence:** 3

**Summary:**

This paper proposes a TD-based reinforcement learning algorithm to efficiently optimize an expressive diffusion or flow-matching policy with respect to a parameterized value function. While the effectiveness is validated, there is a lack of theoretical proof for the proposed approach, and the experiments are not comprehensive.

**Strengths:**

1. Novel algorithm
2. Effectiveness validated

**Weaknesses:**

1. Lack of theoretical proof
2. Incomprehensive experimental setting

**Questions:**

1. The paper lacks a theoretical proof to demonstrate the stability of the proposed method and to explain why it outperforms existing methods.

2. It is recommended to further demonstrate the applicability of the proposed method on tasks beyond long-horizon tasks, such as the MuJoCo tasks in D4RL.

3. A comparison with more state-of-the-art offline-to-online DRL methods should be included in the experiments.

4. How does the performance of the proposed method vary with changes in the number of critics, and does the number of critics affect Q-value evaluation?

5. The proposed method includes many components. It is recommended to analyze the time and space complexity of the proposed method, and to compare its throughput and memory usage with baseline methods in the experiments to assess practical hardware efficiency.

6. How does the performance of the proposed method change as the offline dataset size decreases and its quality declines?

---

> ### Author Response · Authors · 2025-11-21
>
> Thanks for your review and your review has already resulted in various improvements to our paper! We are glad to hear that you find our method novel and effective.
>
> For this rebuttal, **(1) we addressed your main concern on the lack of theoretical proof by providing a concrete theoretical result that shows the critical point of our objective coincides with the optimal behavior-regularized policy; (2) we addressed your concern about experimental comprehensiveness by evaluating our method on 9x more tasks with 4 more seeds and comparing  with 6 new baselines (including 2 DRL baselines) under a rigorous and fair hyperparameter tuning and evaluation protocol, and then by showing that our method achieves the strongest overall performance; (3) we additionally included more comprehensive sensitivity analysis with respect to the critic ensemble size and dataset quality.**
>
> ## 1. Lack of theoretical proof:
>
> We added a new theoretical justification (Proposition 1 on Page 7) showing that **the critical point of the adjoint-matching loss coincides with the optimal behavior-regularized policy**:
>
> $$ \pi^\star \propto \pi_\beta(\cdot \mid s) e^{\tau Q(s, a)}. $$
>
> This is a significant result because it implies that **if the adjoint-matching objective is optimized to convergence (ignoring optimization noise and model mismatch), the resulting flow-matching policy exactly recovers the optimal behavior-regularized policy**.
>
> ---
>
> ## 2. Comparison with SOTA DRL.
>
> For this rebuttal, we additionally compare two strong diffusion RL baselines, QSM [1] and DAC [2] and include the results in Table 1 and Figure 2. In general, while QSM and DAC are able to achieve reasonable performance both offline and online fine-tuning, QAM consistently outperforms them (e.g., QAM achieves an aggregated score of 38 vs. QSM: 32, DAC: 26).
>
> We attach the updated offline RL results below for convenience.
>
> |Category | Method | Aggregated Score (50 tasks) |
> |-|-|-|
> | Gaussian | ReBRAC | 34 |
> | Backprop. | FBRAC | 9 |
> | Backprop. | BAM | 11 |
> | Backprop. | FQL | 34 |
> | Adv. Weighted | FAWAC | 8 |
> | Guidance | CGQL | 27 |
> | Guidance | CGQL-MSE | 33 |
> | Guidance | CGQL-Linex | 34 |
> | Guidance | DAC | 26 |
> | Guidance | QSM | 32 |
> | Post-processing | DSRL | 32 |
> | Post-processing | FEdit | 28 |
> | Post-processing | IFQL | 29 |
> | Adjoint Matching (ours) | QAM | 38 |
> | Adjoint Matching (ours) | QAM-FQL | **41** |
> | Adjoint Matching (ours) | QAM-EDIT | **42** |
>
> The hyperparameter tuning range for QSM and DQC is available in Table 5 and the domain-specific hyperparameters we pick are available in Table 4. We also include the training speed and the parameter count in Table 6. Furthermore, we include a detailed description of these baselines in Appendix C.1. We tune these baselines on two tasks per domain over the hyperparameter tuning range specified in Table 5 with 4 seeds and then pick the best-performing hyperparameter configuration to evaluate them on all 5 tasks for each domain.
>
> [1] Psenka, Michael, et al. "Learning a diffusion model policy from rewards via q-score matching." arXiv preprint arXiv:2312.11752 (2023).
>
> [2] Fang, Linjiajie, et al. "Diffusion actor-critic: Formulating constrained policy iteration as diffusion noise regression for offline reinforcement learning." arXiv preprint arXiv:2405.20555 (2024).
>
> ---
>
> ## 3. How does the performance change with the number of critics?
>
> We include a new ablation experiment in Figure 3 where we show that using an ensemble size of 10 is crucial for good performance of our method.
>
> ---
>
> ## 4. Time and space complexity of the proposed method compared to baseline methods.
>
> We added more details on training speed and parameter count in Table 6. Compared to the strongest diffusion/flow RL method (e.g., QSM, DAC) that does not use distillation, QAM requires ~30% more training time. Compared to the best prior flow method (FQL), QAM requires ~60% more training time. In terms of parameter count, QAM uses <15% more parameters compared to all our baselines.
>
> ---
>
> EDIT (11/24/2025): The experiments have been completed for all seeds. We have updated the table above to reflect the results with 8 seeds.

---

> > ### Comment · Reviewer_V3Vy · 2025-11-25
> >
> > I have read the rebuttal and it addressed most of my concerns. Based on the overall evaluation and other reviewers' comments, I will raise the score a bit.

---

> > > ### Author Response · Authors · 2025-11-26
> > >
> > > Thank you for raising the score and again for your review. Please let us know if there are any remaining concerns or questions that we can address!!

---

> ### Author Response · Authors · 2025-11-21
>
> ## 5. How does the performance of the proposed method change as the offline dataset size decreases and its quality declines?
>
> To better understand how offline data size and quality affects performance, we conducted two additional sets of experiments.
>
> *I. Robustness to noisy offline data*
>
> Our first set of experiments use two of the hardest tasks in the benchmark together with the noisy-style datasets from OGBench. The noisy-style datasets were collected by expert policies with larger, uncorrelated Gaussian noise, resulting in broader state coverage but larger learning challenge. We also evaluated the strongest baselines (FEdit, FQL, ReBRAC) on these noisy datasets. Results are shown Figure 6 with the performance at 250K environment steps summarized in the table below (6 seeds). **Our method exhibits strong robustness to action noises: while baselines collapse almost completely on the noisy datasets, QAM only experiences a modest performance drop.**
>
> | Method  | cube-double-task4 (play) | cube-double-task4 (noisy) | cube-triple-task4 (play) | cube-triple-task4 (noisy) |
> |-|-|-|-|-|
> | ReBRAC | 85[77,93] | 4[0,10] | 1[0,1] | 0[0,0] |
> | FEdit | 98[97,99] | 2[1,3] | 1[1,2] | 0[0,1] |
> | FQL | 96[92,99] | 0[0,0] | 32[26,37] | 0[0,0] |
> | QAM | 95[90,98] | 76[40,99] | 70[59,80] | 52[42,64] |
>
> ---
>
> **EDIT: 11/24/2025 -- we managed to run the same analysis on 14x more tasks (see Figure 7 in our updated PDF). While the overall trend stays the same, we would like to highlight that all baselines fail completely on cube-triple environments (across all 5 tasks) and QAM only experiences some performance drop.**
>
> ---
>
> *II. Sensitivity to dataset size*
>
> The second set of experiments focuses on the dataset size. We use cube-quadruple-task2, where our original experiments used a 100M-size dataset. In this set of experiments, we study how well our method performs when subject to a 75M, 50M, 25M-size dataset for learning (Figure 5). QAM maintains strong performance with half the original dataset (50M), but fails when trained on only 25% of the data.
>
> ---
>
> Thanks again for your review as especially your review has already resulted in various improvements to our paper. **If we have successfully addressed all your concerns, could you kindly update the score?** Please also let us know if you have any additional questions or concerns regarding our paper!

---

### Official Review · Reviewer_VcZG · 2025-10-31

**Soundness:** 2
**Presentation:** 3
**Contribution:** 2
**Rating:** 2
**Confidence:** 4

**Summary:**

The paper introduces Q-Learning with Adjoint Matching (QAM), an actor-critic algorithm with flow-matching policies that uses adjoint matching (a recent technique from generative flow modeling). The paper tackles the problem of optimization instability in backpropagating the critic through multi-step denoising processes in flow or diffusion models. Previous methods either discarded the action gradient and solely relied on action values or only used one-step approximations. QAM uses adjoint matching (AM) to incorporate the critic’s action gradient in training a flow policy subject to a prior constraint. QAM assumes access to a fixed behavior (prior) flow policy and then learns a residual flow policy, which is optimized towards the AM objective. Leveraging the convergence guarantees of AM, the paper claims this converges to the optimal prior regularized policy. The experiments demonstrate QAM has lower hyperparameter sensitivity and better performance compared to many baselines in offline and offline-to-online settings on various OGBench domains.

**Strengths:**

The paper is well-motivated: the problem being addressed is clear, and there are clear arguments for using AM.

The experiments compare to many different algorithms.

**Weaknesses:**

The theoretical guarantees are not detailed, and the word convergence only appears in the intro and conclusion. The derivation relies strongly on SOC objectives from Adjoint Matching, and a clear motivation of the formulation in RL setting, with clearly stated assumptions can make the paper self-contained.

The experiments are quite comprehensive in scope, with many algorithms, which is laudable. However, there are currently three key issues to address.

The first is around how confidence intervals are computed. To quote:
“We report the means and the 95% bootstrapped confidence intervals are computed over 4 seeds”
This is an issue. You cannot get bootstrap confidence intervals from 4 seeds. The number of seeds should be increased, or CIs should only be reported using aggregate performance across environments. I understand experiments can be expensive, however, it is key to only make claims supported by the evidence. It is possible to only show qualitative behavior, such as individual runs in an environment (with 4 lines), and then make stronger claims aggregating across environments. Or, of course, the number of runs (seeds) can be increased.

The second key concern is how hyperparameters were chosen. Appendix D lists the hyperparameters and states that they needed to be tuned for each domain. How was this done? When tuning per environment, it is key to control for the number each method gets to tune, and in general can be difficult to keep fair. Tuning hypers can be very misleading in terms of the actual utility of an algorithm, because different settings of hypers can essentially give you a different algorithm. Ideally, it is better to tune across environments or at least explain why the current tuning is valid and not misleading.

The third key concern is a missing baseline (in Table 1): a strong offline RL method that is not focused on diffusion policies. RLPD is used later, which just uses a Gaussian, but it is not an offline method. In general, for both the offline setting and the offline-online setting, it could be more informative to consider IQL. Note also that one baseline IFQL uses IQL, so it is sensible to include for this reason too. It is a strong good baseline for the offline experiments. And, for offline-online, it is also likely a stronger baseline that RLPD, since RLPD does not make as much use of the offline data as IQL. I think it is key to include IQL, or another method that has been shown to perform well in offline and offline-online, such as In-sample Actor Critic (see https://arxiv.org/abs/2302.14372). This method has an update that is a lot like SAC with a small modification to handle out-of-distribution actions.


(Putting Minor Comments here, because there is not separate box for them. These Minor Comments are not big weaknesses.)
1. “more convenient to directly use the same offline RL objective for both offline pre-training and online fine-tuning” Why is it more convenient? There are stronger reasons for doing so and should be listed here.

2. The paragraph at the beginning of Section 2 is too long and should be split up into multiple ideas. Long paragraphs throughout should be broken into multiple, with clear topic sentences, for readability and clarity.

3. It is a bit confusing to use v and r in the formulas for adjoint matching, even if that is their original terminology, due to the clash with value functions in RL and the fact that you will use q rather than r. You could simply use q right away, and consider a different variable for v

4. The sensitivity analysis is highly appreciated, but there could be just a little bit more there. In the sensitivity analysis in Figure 4, there is not a lot of difference between the values of tau nor without gradient clipping. This suggests more values of tau should be tested to truly see when there is failure. This is not critical, but would make this result more useful.

**Questions:**

Questions
1. How do you plan to address the issue with CIs based on 4 seeds?
2. Can you explain how hyperparameters were tuned? (see above)
3. Is it possible to run an experiment compared to a stronger offline method, like IQL or Insample AC, with a Gaussian policy? Or can you explain why you do not think this is needed?

The current decision is based on these omissions, but I am willing to adjust my score based on the responses from the authors. The current scores reflect uncertainty on the outcomes, and Soundness and Contribution would particularly be increased when addressing these omissions.

---

> ### Author Response · Authors · 2025-11-21
>
> Thanks for your constructive feedback and detailed comments. We really appreciated it! Regardless of the outcome of the review process, your review has already improved the quality of our paper by a lot!
>
> We addressed **1) your concerns on lack of theoretical guarantees by including a new theoretical result that shows the critical point of our loss function coincides with the optimal behavior-regularized policy, and 2) your concerns on empirical results by (i) doubling the number of seeds (4 => 8, some experiments are still running) and (ii) 10x the number of tasks (5 => 50) and (iii) your concern on missing baseline by additionally comparing with a strong offline RL Gaussian baseline (ReBRAC).**
>
> Now we detail how we addressed them below with all changes reflected in the latest PDF of our paper in blue.
>
> ---
>
> ## 1. Lack of theoretical analysis
>
> We added a new theoretical justification (Proposition 1 on Page 7) showing that **the critical point of the adjoint-matching loss coincides with the optimal behavior-regularized policy**:
>
> $$ \pi^\star \propto \pi_\beta(\cdot \mid s) e^{\tau Q(s, a)}. $$
>
> It implies that **if the adjoint-matching objective is optimized to convergence (ignoring optimization noise and model mismatch), the resulting flow-matching policy exactly recovers the optimal behavior-regularized policy**.
>
> ---
>
> ## 2. Lack of seeds:
>
> Thanks for pointing out this issue and this is indeed a very valid concern! For the rebuttal, we ran 4 additional seeds (8 seeds in total) and evaluated on 9 times more tasks (50 tasks across 10 OGBench domains). The updated offline results are summarized in the markdown table below (also Table 1 in our updated PDF for individual domain breakdown). We also include our updated online fine-tuning results in Figure 2. For each online fine-tuning result subplot, we aggregate over 5 tasks. Note that this is different from the plots in our original submission where we only had 1 task per plot.
>
> |Category | Method | Aggregated Score (50 tasks) |
> |-|-|-|
> | Gaussian | ReBRAC | 34 |
> | Backprop. | FBRAC | 9 |
> | Backprop. | BAM | 11 |
> | Backprop. | FQL | 34 |
> | Adv. Weighted | FAWAC | 8 |
> | Guidance | CGQL | 27 |
> | Guidance | CGQL-MSE | 33 |
> | Guidance | CGQL-Linex | 34 |
> | Guidance | DAC | 26 |
> | Guidance | QSM | 32 |
> | Post-processing | DSRL | 32 |
> | Post-processing | FEdit | 28 |
> | Post-processing | IFQL | 29 |
> | Adjoint Matching (ours) | QAM | 38 |
> | Adjoint Matching (ours) | QAM-FQL | **41** |
> | Adjoint Matching (ours) | QAM-EDIT | **42** |
>
> **Overall, our main conclusion remains the same so far: QAM continues to achieve the strongest offline RL and online fine-tuning performance compared to all prior approaches (e.g., with an aggregated offline score of 38 compared to the previous best score of 34)**.
>
> Furthermore, **we experimented with two variants of QAM where we combined FQL/FEdit on top of the QAM-finetuned policy and showed that either of them can improve upon QAM’s performance even further (e.g., offline aggregated score improving from 38 to 41 [+FQL] and 42 [+FEdit] respectively).** We included the descriptions of these two variants towards the end of Section 4 (starting from the bottom of page 7).
>
> For online fine-tuning experiments, we use the QAM-FEdit variant of our method and we use CGQL-Linex variant of CGQL due to their good performance in the offline experiments. **Our method outperforms all prior methods significantly on the cube-triple domain and is the most robust method across the board.** For example, compared to QAM, QSM fine-tunes better on antmaze-large and giant, but struggles on all other tasks except on cube-double, where it performs similarly to our method. FQL fine-tunes slightly better on antmaze-giant and much slower on both puzzle-4x4 and cube-triple. ReBRAC starts off with good offline pre-training performance in the beginning of online fine-tuning on antmaze-large and giant, but it fails to fine-tune effectively on giant. It also completely fails on puzzle-4x4.
>
> Since online fine-tuning experiments are generally more expensive, we stick with the original 5 domains that we evaluated on during our initial submission but now with 5 tasks for each domain, and additionally included results for the harder puzzle task to cover all different task characteristics (e.g., locomotion/navigation: antmaze, manipulation reaching: puzzle, manipulation pick and place: cube). We also cut down the number of online environment steps (from 500K to 250K) to cut down the computational cost as we have found that the majority of learning happens within the first 200K steps.
>
> ---
>
> EDIT (11/24/2025): The experiments have been completed for all seeds. We have updated the table above to reflect the results with 8 seeds.

---

> ### Author Response · Authors · 2025-11-21
>
> ## 3. How are hyperparameters tuned?
>
> Thanks for this clarification question! For our original submission, we tuned directly on the tasks we evaluated via a grid search over a set of parameters for both our method and our baselines (see Table 5 in our original submission). For all results in the rebuttal, since we include more tasks for each domain, we retuned all the baselines and methods on two representative tasks per domain (task1 and 4 for locomotion domains, and task 2 and 4 for manipulation domains), and then used the same hyperparameters for all five tasks in the same domain. We used task1 for locomotion and task2 for manipulation domains as they are the default/representative task recommended by OGBench. We additionally tuned on task4 because we have found in our experiments that the performance of a combination of task4 and the default task is more representative. We used 4 seeds (1001, 2002, 3003, 4004) for tuning on the two of the five tasks, and used 8 different seeds (10001, 20002, 30003, 40004, 50005, 60006, 70007, 80008) for evaluations on all five tasks for each domain. For the updated PDF, the hyperparameter tuning ranges are in Table 5 and the domain-specific hyperparameters for each method are in Table 4.
>
> ---
>
> ## 4. Lack of strong Gaussian policy baseline.
>
> Thanks for pointing it out. For this rebuttal we included results for a strong Gaussian policy baseline, ReBRAC. The reason we did not compare with Gaussian baselines was because they were previously reported to underperform flow/diffusion baselines in many recent papers (e.g., [1, 2, 3]). However, inspired by your comments and feedback, we decided to include an additional Gaussian baseline in our paper.
>
> We tried and tuned three Gaussian methods, ReBRAC [4], IQL [5], Insample AC [6]. From our tuning experiments, we have found ReBRAC to perform the best so we picked it as the Gaussian baseline that we ran in our rebuttal experiments. We did not include the results for the IQL and Insample AC because we are not fully confident that we have properly debugged/tuned them on OGBench. For the remaining duration of the rebuttal, we will keep working on them and will update the thread once we have the results that we are confident in.
>
> For ReRBAC, we tuned two key parameters which we found to matter the most in practice. The first parameter was the behavior cloning coefficient and the second parameter was the policy noise magnitude (for both backup and the policy evaluations). The parameter range for each of them and the domain-specific values are detailed in Table 5 and Table 4 respectively.
>
> In terms of performance, we actually found ReBRAC to be quite strong on locomotion domains. It achieves the highest score on both al (antmaze-large) and ag (antmaze-giant) offline. Aggregated over all 10 domains, 50 tasks, it matches the prior best flow method, FQL, with an aggregated score of 34. For online fine-tuning performance, ReBRAC generally performs worse than the other methods except on the two antmaze domains (see Figure 2). While we were a bit surprised by how good ReBRAC performs in the offline phase (as prior work often reported ReBRAC to perform worse), we are confident that the comparison is fair to the best of our capability as we tried our best to put in a similar amount of effort (e.g., number of configurations in the tuning sweep) in tuning both ReBRAC and other baselines. We hope these experiments serve as a more comprehensive comparison between Gaussian and flow/diffusion baselines for the research community.
>
> **Overall, our approaches still outperform all prior methods across the board, with an average score of 38 (vs. prior best of 34). In addition, as we mentioned above, we could further push the performance of our method by combining it with FQL and FEdit, achieving an aggregated score of 41 and 42 respectively.**
>
> [1] Wang, Zhendong, Jonathan J. Hunt, and Mingyuan Zhou. "Diffusion policies as an expressive policy class for offline reinforcement learning." arXiv preprint arXiv:2208.06193 (2022).
>
> [2] Chen, Huayu, et al. "Score regularized policy optimization through diffusion behavior." arXiv preprint arXiv:2310.07297 (2023).
>
> [3] Park, Seohong, Qiyang Li, and Sergey Levine. "Flow q-learning." arXiv preprint arXiv:2502.02538 (2025).
>
> [4] Tarasov, Denis, et al. "Revisiting the minimalist approach to offline reinforcement learning." Advances in Neural Information Processing Systems 36 (2023): 11592-11620.
>
> [5] Kostrikov, Ilya, Ashvin Nair, and Sergey Levine. "Offline reinforcement learning with implicit q-learning." arXiv preprint arXiv:2110.06169 (2021).
>
> [6] Zhang, Hongchang, et al. "In-sample actor critic for offline reinforcement learning." The Eleventh International Conference on Learning Representations. 2023.
>
> ---
>
> EDIT (11/24/2025): The experiments have been completed for all seeds. We have updated the aggregated score for QAM-FQL (from 40 to 41) above to reflect the results with 8 seeds.

---

> > ### Author Response · Authors · 2025-11-21
> >
> > ## 5. Additional sensitivity analysis
> >
> > We expanded our sensitivity analysis, studying the influence of following components in our algorithm (Figure 3):
> > - pessimistic backup coefficient ($\rho$),
> > - gradient clipping,
> > - number of flow steps ($T$),
> > - critic ensemble size ($K$), and finally
> > - temperature coefficient ($\tau$), with a larger range as requested by the reviewer.
> >
> > From our analysis, we found that all of these components contribute to QAM's effectiveness. Among them, the pessimistic backup coefficient, and the temperature parameter ($\tau$) have the biggest impact on QAM’s performance and need to be tuned. In our experiments we use $\tau=0$ for humanoidmaze domains and $\tau=0.5$ for other domains.
> >
> > ---
> >
> > ## 6. Writing feedback:
> >
> > Thanks for your writing feedback! We revised the related work section according to your comments. Regarding the notations for $v$ and $r$, we have changed the $v$’s and $r$’s to $f$ and $Q$.
> >
> > ---
> >
> > Thanks again for your review. Your review has already resulted in many significant improvements to our paper. **If we have successfully addressed all your concerns, could you kindly update the score?** Please also let us know if you have any additional questions or concerns regarding our paper!

---

> > > ### Author Response · Authors · 2025-11-28
> > > **A quick follow-up**
> > >
> > > As the end of the rebuttal discussion period is approaching quickly, we would like to post a short follow-up to check whether our rebuttals have addressed all your concerns. If there are any remaining questions or concerns, please let us know! Thanks again for taking the time to review our paper and we really appreciate your review as it has significantly improved our work!

---

### Official Review · Reviewer_8dDG · 2025-11-03

**Soundness:** 3
**Presentation:** 2
**Contribution:** 3
**Rating:** 6
**Confidence:** 2

**Summary:**

The paper addresses the instability that arises when backpropagating policy gradients through action flows, which often leads to practical difficulties when optimizing a policy with a learned Q-network. To tackle this issue, the authors employ a recent technique called adjoint matching. Rather than training a single flow-matching policy solely on an offline dataset, they utilize the behavior policy to generate adjoint states that provide direct step-to-step supervision signals for the residual policy. This results in a stable adjoint-matching objective that enables local supervision over the residual flow while still recovering an optimal prior-regularized policy. The approach is compatible with settings where a prior policy is provided, and the authors demonstrate its effectiveness through experiments in both offline and offline-to-online scenarios.

**Strengths:**

The paper addresses a critical challenge in flow-matching-based policy learning. While the current setting is limited to cases where a behavior policy is available, the authors introduce a well-grounded formulation that establishes a clear connection to adjoint-matching techniques in generative modeling. The resulting policy maintains strong expressivity, enabling more diverse action distributions to be represented. I believe this capability can offer benefits to reinforcement learning that extend beyond the specific experimental settings evaluated in the paper.

**Weaknesses:**

The method relies heavily on accurate action gradients, though this dependence is common across many RL algorithms. The authors address this challenge in practice, for example by training a critic ensemble for more reliable gradient estimates. That said, there may be further opportunities to explore more advanced techniques to improve gradient accuracy.

The evaluation is limited to OGbench domains. Including results on benchmarks with more diverse characteristics would further strengthen the paper’s contributions. For example, demonstrating performance across policies of varying quality could offer additional insights into the method’s robustness and general applicability, dense rewards, offline datasets of different qualities, etc. If an offline dataset is collected using a policy with low behavioral diversity, it is unclear how this would impact the effectiveness of the proposed approach. A discussion or empirical analysis on this aspect would be valuable.

**Questions:**

I am also interested in the computational cost of the method—approximately how many GPU hours are required to train a policy? Also, is there any observed numerical instabilities for high dimensionality?

---

> ### Author Response · Authors · 2025-11-21
>
> Thank you for taking the time to review our paper and we appreciate your positive review! We are glad that you believe our method can offer benefits to RL beyond the settings evaluated in the paper and it is very encouraging for us!
>
> To address your concerns, **we conducted additional experiments demonstrating that QAM is reasonably robust to data corruptions, and we also provided details on the computational cost for both our method and all baselines.**
>
> ## 1. Lack of analysis on offline data with different qualities
>
> To better understand how offline data quality affects performance, we conducted two additional sets of experiments.
>
> *I. Robustness to noisy offline data*
>
> Our first set of experiments use two of the hardest tasks in the benchmark together with the noisy-style datasets from OGBench. The noisy-style datasets were collected by expert policies with larger, uncorrelated Gaussian noise, resulting in broader state coverage but larger learning challenge.
>
> We evaluated QAM alongside the strongest baselines (FEdit, FQL, ReBRAC). Results are shown in Figure 6 with the performance at 250K environment steps summarized in the table below (6 seeds). **Our method exhibits strong robustness to action noises: while baselines collapse almost completely on the noisy datasets, QAM only experiences a modest performance drop.**
>
> | Method  | cube-double-task4 (play) | cube-double-task4 (noisy) | cube-triple-task4 (play) | cube-triple-task4 (noisy) |
> |-|-|-|-|-|
> | ReBRAC | 85[77,93] | 4[0,10] | 1[0,1] | 0[0,0] |
> | FEdit | 98[97,99] | 2[1,3] | 1[1,2] | 0[0,1] |
> | FQL | 96[92,99] | 0[0,0] | 32[26,37] | 0[0,0] |
> | QAM | 95[90,98] | 76[40,99] | 70[59,80] | 52[42,64] |
>
> ---
>
> **EDIT: 11/24/2025 -- we managed to run the same analysis on 14x more tasks (see Figure 7 in our updated PDF). While the overall trend stays the same, we would like to highlight that all baselines fail completely on cube-triple environments (across all 5 tasks) and QAM only experiences some performance drop.**
>
> ---
>
>
> *II. Sensitivity to dataset size*
>
> The second set of experiments focuses on the dataset size. We use cube-quadruple-task2, where our original experiments used a 100M-size dataset. In this set of experiments, we study how well our method performs when subject to a 75M, 50M, 25M-size dataset for learning (Figure 5). QAM maintains strong performance with half the original dataset (50M), but fails when trained on only 25% of the data.
>
> ## 2. Computational cost of the method
>
> We added more details on training speed in Table 6. Compared to the strongest diffusion/flow RL method (e.g., QSM, DAC) that does not use distillation, QAM requires ~30% more training time. Compared to the best prior flow method (FQL), QAM requires ~60% more training time.
>
> For 1M-step offline training, it takes ~1.6 hours (excluding evaluation time).
>
> ## 3. Do you observe numerical instabilities for high dimensionality?
>
> For this rebuttal, we additionally evaluate on two humanoidmaze domains, which have the highest action dimensionality (21) among all tasks evaluated. We did not observed any numerical instability so far on these domains. QAM performs on par with FQL (the strongest flow method) on both humanoidmaze domains (see Table 1).
>
> Thanks again for your review. Your review has already resulted in various improvements to our paper. **Please also let us know if you have any additional questions or concerns regarding our paper!**

---

### Author Response · Authors · 2025-11-21
**General Comments**

We would like to thank all reviewers for the detailed and insightful reviews and area chair for facilitating the review discussion. For the updated version of PDF, we have made several improvements to our empirical evaluations and theoretical analysis with all changes in the PDF are highlighted in blue.

**(A) In response to Reviewer VcZG, we increased the number of tasks (5=>50), seeds (4=>8) for our evaluations.**

**(B) We also included a Gaussian baseline (as requested by Reviewer VcZG), two Diffusion RL baselines (as requested by Reviewer V3Vy), and a “basic adjoint matching” ablation baseline (as requested by YuTk).**

**(C) In response to Reviewer VcZG, Reviewer V3Vy, Reviewer YuTk, we also included a new theoretical result that provides guarantees that the critical point of our adjoint matching objective coincides with the optimal behavior regularized behavior distribution.**


We now description these changes in more detail below:

---

### 1.  Improved empirical evaluations with more tasks (5 => **50 tasks!**), more baselines (**6 new baselines!**) and more seeds (4 => **8 seeds**).

We additionally included 6 new baselines:
- a strong Gaussian baseline (ReBRAC),
- two recent DRL methods (QSM and DQC),
- two improved classifier-guidance-based baselines (CGQL-MSE/Linex), and
- an ablation baseline (BAM: basic adjoint matching).

All baseline descriptions can be found in Section 5 and Appendix C.1.

Our updated offline RL results are summarized in the markdown table below (also Table 1 in our updated PDF). Our updated online fine-tuning results are in Figure 2. For each online fine-tuning result subplot, we aggregate over 5 tasks. Note that this is different from the plots in our original submission where we only had 1 task per plot.

|Category | Method | Aggregated Score (50 tasks) |
|----------|--------|-----|
| Gaussian | ReBRAC | 34 |
| Backprop. | FBRAC | 9 |
| Backprop. | BAM | 11 |
| Backprop. | FQL | 34 |
| Adv. Weighted | FAWAC | 8 |
| Guidance | CGQL | 27 |
| Guidance | CGQL-MSE | 33 |
| Guidance | CGQL-Linex | 34 |
| Guidance | DAC | 26 |
| Guidance | QSM | 32 |
| Post-processing | DSRL | 32 |
| Post-processing | FEdit | 28 |
| Post-processing | IFQL | 29 |
| Adjoint Matching (ours) | QAM | 38 |
| Adjoint Matching (ours) | QAM-FQL | **41** |
| Adjoint Matching (ours) | QAM-EDIT | **42** |

(note: we bold all the entries as long as its average fall within the best method’s confidence interval)

To ensure a fair evaluation, we used a **rigorous hyperparameter tuning protocol**:
- For each method, we tuned hyperparameters using four seeds (1001, 2002, 3003, 4004) on two of the 5 tasks per domain, using comparable hyperparameter search set cardinalities to mitigate over-tuning bias (see Table 5 for the hyperparameter sweep ranges).
- We then evaluated the best hyperparameter configuration using 8 new seeds (10001, 20002, 30003, 40004, 50005, 60006, 70007, 80008) across all five tasks in each domain (see Table 4 for the domain-specific hyperparameters)

**Overall, our main conclusion remains the same: QAM continues to achieve the strongest offline RL and online fine-tuning performance compared to all prior approaches (e.g., with an aggregated offline score of 38 compared to the previous best score of 34)**.

Furthermore, **we experimented with two variants of QAM where we combined FQL/FEdit on top of the QAM-finetuned policy and showed that either of them can improve upon QAM’s performance even further (e.g., offline aggregated score improving from 38 to 41 [+FQL] and 42 [+FEdit] respectively).** We included the descriptions of these two variants towards the end of Section 4 (starting from the bottom of page 7).

---

EDIT (11/24/2025): The experiments have been completed for all seeds. We have updated the table above to reflect the results with 8 seeds.

---

> ### Author Response · Authors · 2025-11-21
>
> ### 2. New ablation experiments on basic adjoint matching
>
> To study the stability benefits of our adjoint-matching objective, we added an ablation study comparing QAM with the basic adjoint matching (BAM) objective (in Eq. 12). BAM admits the same optimal solution as QAM but does not use “lean adjoint” states, and thus contains extra terms that incurs additional variance during training (see a more comprehensive discussion in Section 5.2 of [1]).
>
> As we expected, **QAM significantly outperforms BAM across all tasks, providing evidence that lean-adjoint is crucial for performance**.
>
> [1] Domingo-Enrich, Carles, et al. "Adjoint matching: Fine-tuning flow and diffusion generative models with memoryless stochastic optimal control." arXiv preprint arXiv:2409.08861 (2024).
>
> ---
>
> ### 3. More comprehensive sensitivity analyses
>
> We expanded our sensitivity analysis, studying the influence of following components in our algorithm:
> - pessimistic backup coefficient ($\rho$),
> - gradient clipping,
> - number of flow steps ($T$),
> - critic ensemble size ($K$), and final
> - temperature coefficient ($\tau$)
>
> From our analysis, we found that all of these components contribute to QAM's effectiveness. Among them, the pessimistic backup coefficient, and the temperature parameter ($\tau$) have the biggest impact on QAM’s performance and need to be tuned. In our experiments we use $\tau=0$ for humanoidmaze domains and $\tau=0.5$ for other domains.
>
> ---
>
> ### 4. New theoretical result
>
> We added a new theoretical justification (Proposition 1 on Page 7) showing that **the critical point of the adjoint-matching loss coincides with the optimal behavior-regularized policy**:
>
> $$ \pi^\star \propto \pi_\beta(\cdot \mid s) e^{\tau Q(s, a)}. $$
>
> This is a significant result because it implies that **if the adjoint-matching objective is optimized to convergence (ignoring optimization noise and model mismatch), the resulting flow-matching policy exactly recovers the optimal behavior-regularized policy**.
>
> ---
>
> **We again thank all reviewers and AC and please let us know if there are any additional questions or concerns regarding our paper!**

---

### Meta-Review · Area_Chair_fKss · 2026-01-08

**Summary:**

This meta-review summarizes the review process and justifies the acceptance decision for the paper "Q-Learning with Adjoint Matching (QAM)". The decision is based on the authors' comprehensive rebuttal, which addressed the majority of the reviewers' concerns.

The reviewers acknowledged the paper's novel contribution in stabilizing gradient-based optimization for expressive flow/diffusion policies via adjoint matching. Key concerns that informed the initial scores included:

Lack of formal guarantees that the method converges to the optimal policy.

Limited scale (only 5 tasks, 4 seeds), unclear hyperparameter tuning protocols, and missing strong baselines.

Experiments were confined to OGBench; generalizability to other task types (e.g., dense-reward domains) and data qualities was unclear.

**Reviewer Concerns:**

Addressed:

Added Proposition 1, proving that the critical point of the adjoint-matching objective coincides with the optimal behavior-regularized policy.

Expanded experiments from 5 to 50 tasks and from 4 to 8 seeds. Implemented a rigorous hyperparameter tuning protocol (tuned on two tasks per domain, evaluated on five with separate seeds) to ensure fairness.

Added a strong Gaussian baseline (ReBRAC), two diffusion RL baselines (QSM, DAC), improved classifier-guidance variants (CGQL-MSE/Linex), and an ablation baseline (Basic Adjoint Matching).

Outstanding:
The experiments remain within OGBench, though the benchmark itself includes diverse task characteristics.

**Reviewer Scores:**

Reviewer 8dDG (initial 6): Concerns about data quality and computational cost were addressed. The reviewer would likely maintain a score of 6.

Reviewer VcZG (initial 2): All major concerns (theoretical, empirical scale, hyperparameter tuning, missing baseline) were addressed with new theory, expanded experiments, and the addition of ReBRAC. Had the reviewer been able to participate, they would likely have raised their score to 6.

Reviewer V3Vy (initial 4): Explicitly raised score to 6 after the rebuttal, citing addressed concerns on theory, baselines, and sensitivity.

Reviewer YuTk (initial 4): Explicitly raised score to 6 after the rebuttal, acknowledging improvements in theory, experiments, and ablation studies.

---

### Decision · Program_Chairs · 2026-01-26

Accept (Poster)